# In situ photodeposition of platinum clusters on a covalent organic framework for photocatalytic hydrogen production

Yimeng Li[1,3], Li Yang[1,3], Huijie He[1], Lei Sun 🄳 [1], Honglei Wang[1], Xu Fang[1], Yanliang Zhao[1], Daoyuan Zheng[1], Yu Qi[2], Zhen Li 🄳 [1✉] & Weiqiao Deng 🄳 [1,2✉]

Photocatalytic hydrogen production has been considered a promising approach to obtain green hydrogen energy. Crystalline porous materials have arisen as key photocatalysts for efficient hydrogen production. Here, we report a strategy to in situ photodeposit platinum clusters as cocatalyst on a covalent organic framework, which makes it an efficient photocatalyst for light-driven hydrogen evolution. Periodically dispersed adsorption sites of platinum species are constructed by introducing adjacent hydroxyl group and imine-N in the region of the covalent organic framework structural unit where photogenerated electrons converge, leading to the in situ reduction of the adsorbed platinum species into metal clusters by photogenerated electrons. The widespread platinum clusters on the covalent organic framework expose large active surface and greatly facilitate the electron transfer, finally contributing to a high photocatalytic hydrogen evolution rate of 42432 $\mu$mol g$^{-1}$ h$^{-1}$ at 1 wt% platinum loading. This work provides a direction for structural design on covalent organic frameworks to precisely manipulate cocatalyst morphologies and positions at the atomic level for developing efficient photocatalysts.

---

[1] Institute of Molecular Sciences and Engineering, Institute of Frontier and Interdisciplinary Science, Shandong University, Qingdao, Shandong 266237, China. [2] Dalian National Laboratory for Clean Energy, Dalian Institute of Chemical Physics, Chinese Academy of Sciences, Dalian 116023, China. [3] These authors contributed equally: Yimeng Li, Li Yang. ✉email: zhen_li@sdu.edu.cn; dengwq@sdu.edu.cn

Converting renewable solar energy into clean and carbon neutral fuels is desirable for reliving the current energy and environmental crisis[1,2]. Photocatalytic hydrogen production from water splitting directly harvest solar light and produce hydrogen in a simple device and has been regarded as an economical technology for produce green energy sources[3]. Among the various photocatalysts investigated so far[4,5], porous materials with nanoscale pore structure and ultra-high surface area have been favored by researchers by providing larger active spaces for the adsorption of reactants and the subsequent catalytic reactions[6–8]. Covalent organic frameworks (COFs), a class of crystalline porous polymers built by covalent bonds between various ligands, have shown great advantages in photocatalytic research in recent years[9]. The entire covalent bond linkage makes COFs possess excellent chemical stabilities, especially in imine-linked and other nitrogen-containing COFs[10]. As it is possible to atomically precise integration of organic units to create predesigned and various skeletons, COFs have unlimited chemical adjustability and their optical and electrical properties such as light capture, charge separation and charge transport can be modulated. The π-conjugated structure of COFs both in plane and between layers is beneficial to increase the mobility of charge carriers[11]. Moreover, compared with traditional molecular catalysts, COFs lock the photoactive building units in a rigid structure to prevent photocorrosion and improve the life of the excited state[12,13].

In 2014, Lotsch et al.[14] constructed a water-stable hydrazone bond group TFPT-COF and first applied it to photocatalytic hydrogen evolution. After that, various COF materials were designed and investigated as photocatalysts, and the hydrogen evolution reaction (HER) performance was drastically enhanced. Cooper et al.[15] designed fused-sulfone-COF (FS-COF), which exhibited HER activity as high as 16,300 μmol h$^{-1}$ g$^{-1}$ when dye-sensitized. Recently, Guo and co-workers[16] reported an electron-transfer mediator integrated COF that exhibited a hydrogen evolution rate up to 34,600 μmol h$^{-1}$ g$^{-1}$. Among all the efficient HER systems, the expensive noble-metal platinum was found to be an excellent co-catalyst to catalyse surface proton reduction to hydrogen[17]. However, the rational deposition of Pt on the framework of a COF was rarely considered in the research, resulting in the random growth of Pt nanoparticles of a large size with limited exposed active surface even at a high loading amount. In view of the tunable and regular chemical structure as well as the abundant porous channels[18], COF photocatalysts indeed provide an ideal platform for the precise control of Pt deposition, which promotes deeper understanding of Pt growth and the maximum utilization of expensive noble metals.

Here, we design a type of covalent organic framework photocatalyst (PY-DHBD-COF), which bears an adjacent hydroxyl group and imine bond in each constitutional unit. In light-driven hydrogen evolution, the adsorption of Pt precursor PtCl$_6^{2-}$ on a specific site near hydroxyl group and imine-N and the subsequent photoreduction lead to in situ photodeposition of Pt clusters on the 2D COF surface layer with high dispersion. This well-dispersed deposition contributes to the optimal utilization of Pt cocatalyst and a hydrogen evolution rate up to 42,432 μmol g$^{-1}$ h$^{-1}$ at 1 wt% Pt loading. This work provides a direction for manipulating the deposition of photocatalytic co-catalysts at the atomic level by taking full advantage of the designable and tunable pore structure of a COF to develop an efficient photocatalyst.

## Results

**Synthesis and characterizations of COF.** PY-DHBD-COF was synthesized via a Schiff-base condensation reaction of 1,4-dihydroxybenzidine (DHBD) with 1,3,6,8-tetra(4-formylphenyl)pyrene (PY-CHO) in the presence of acetic acid in a mixed solvent of n-butyl alcohol (n-BuOH) and o-dichlorobenzene (o-DCB) (Fig. 1)[19,20]. Structural models for PY-DHBD-COF with a perfectly eclipsed AA stacking pattern and AB stacking pattern are shown in Fig. 2a and b. The as-synthesized PY-DHBD-COF was analyzed by Fourier transform infrared (FT-IR) spectroscopy (Fig. 2c), and the characteristic signal of C=N stretching emerged at 1628 cm$^{-1}$, along with the disappearance of the aniline N–H stretching band at 3352 cm$^{-1}$. The signal at 1696 cm$^{-1}$ (C=O stretching) could be attributed to the terminal residual group of PY-DHBD-COF[19,21]. The solid-state NMR spectra featured the characteristic signal at 165 ppm, revealing the imino bonds (C=N) of PY-DHBD-COF (Supplementary Fig. 1a). The powder X-ray diffraction (PXRD) results show typical crystalline patterns for PY-DHBD-COF with diffraction peaks at 2.86°, 4.28°, 6.04°, 9.28°, 12.54°, and 23.42°, which are attributed to the (110), (020), (220), (330), (440), and (001) facets, respectively (Fig. 2d)[19]. Pawley refinements show that the diffraction pattern of PY-DHBD-COF was consistent with an orthorhombic lattice (R$_p$ = 5.58%, R$_{wp}$ = 7.74%) with unit cell parameters (a = 39.355(3) Å, b = 41.382(1) Å, c = 3.894(7) Å), which matches well with the idealized AA-stacking model. Moreover, a variety of other possible AA stackings including AA flipped and slipped models were considered, and the simulated XRD patterns are almost same

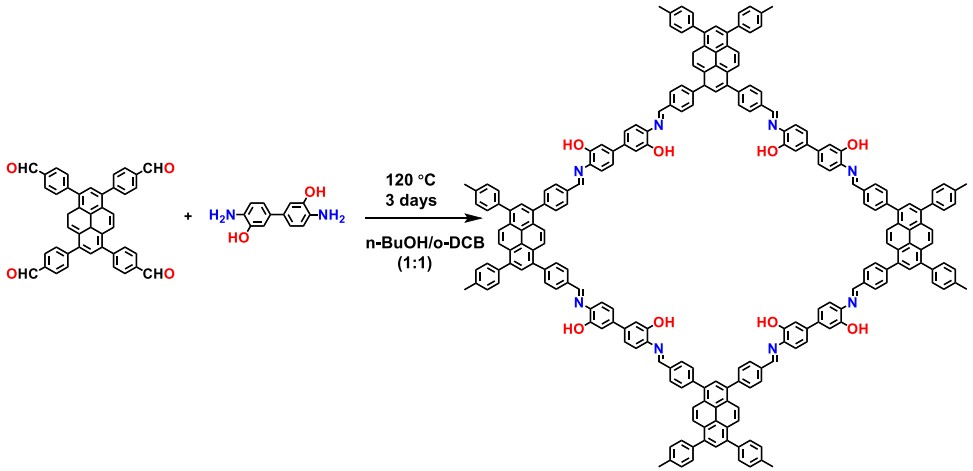

**Fig. 1** Synthesis process and structure of PY-DHBD-COF.

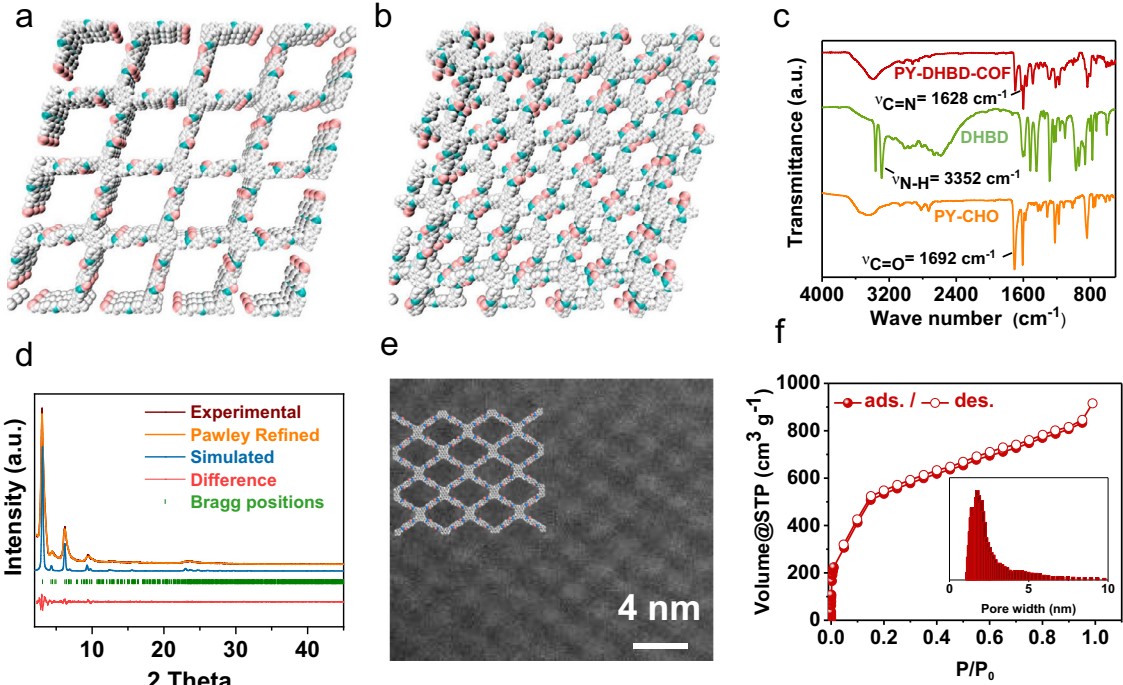

**Fig. 2 Structural characterizations and morphology of PY-DHBD-COF. a** AA-stacked structure and **b** AB-stacked structure of PY-DHBD-COF, the balls in different colors represent different atoms: C, gray; N, green; O, pink. **c** FT-IR spectra of PY-DHBD-COF, and the building monomers. **d** Experimental and simulated PXRD patterns of PY-DHBD-COF. **e** HAADF-STEM image of PY-DHBD-COF. **f** $N_2$ adsorption isotherms of PY-DHBD-COF. Insert: pore size distribution of PY-DHBD-COF.

(Supplementary Fig. 1c), making it difficult to distinguish between these. Therefore, we refer to the idealized, perfectly eclipsed AA stacking model due to its lowest stacking energy (Supplementary Table 1). The scanning electron microscopy (SEM) image of PY-DHBD-COF shows a hollow bar morphology with a diameter of 1 μm (Supplementary Fig. 1b). The high-angle annular dark-field scanning transmission electron microscopy (HAADF-STEM) results (Fig. 2e) of PY-DHBD-COF confirm that the COF has an ordered tetragonal pore structure with a pore size of ~2.5 nm, which is consistent with the in-plane pore channels of 2.7 nm in the proposed AA-stacked COF structure.

The permanent porosity of PY-DHBD-COF was assessed by nitrogen adsorption isotherm measured at 77 K. The adsorption curves bear the characteristic features of type II isotherms (Fig. 2f) with a Brunauer–Emmett–Teller (BET) surface area as high as 1893 m$^2$ g$^{-1}$. Non-local density functional theory (NL-DFT) calculations reveal the pore size distribution of PY-DHBD-COF is centred at 2.5 nm, matching the predicted value (2.7 nm) of the AA-stacked COF structure. The thermal stability and structural stability of PY-DHBD-COF were further investigated. Thermogravimetric analysis (TGA) of PY-DHBD-COF shows that the COFs are thermally stable at high temperatures up to 400 °C (Supplementary Fig. 2). The structural stability was evaluated by soaking COF in different solvents, including water, aqueous acetic acid (3 M), anhydrous tetrahydrofuran, acetone and methanol, and the PXRD crystalline patterns of the treated COF were still maintained, indicating good structural stability (Supplementary Fig. 3).

For the application of PY-DHBD-COF in photocatalysis, the optical and electronic properties were studied first. Ultraviolet-visible (UV-Vis) diffuse reflectance spectra show that the COF absorbs light from the UV region to the visible region with an absorption edge of ~510 nm (Fig. 3a). The optical band gap is calculated as 2.28 eV according to the corresponding Tauc plot. On the basis of ultraviolet photoelectron spectroscopy (UPS)

(Fig. 3b), the relative valence band maximum (VBM) of PY-DHBD-COF is calculated to be 1.29 eV (vs. NHE)[16]. Therefore, the energy of the conduction band minimum (CBM) of PY-DHBD-COF is calculated to be −0.99 eV (vs. NHE).

The electronic properties of PY-DHBD-COF were further investigated by electrochemical testing. The Mott–Schottky plots (Fig. 3c) show that PY-DHBD-COF is an n-type semiconductor with a flat band potential at ~ −0.72 (V vs. NHE); accordingly, the CBM is estimated to be −0.92 (V vs. NHE), which is not much different from the values obtained by UPS[22]. The energy level diagram is described in Fig. 3d. As we all know that in photocatalytic hydrogen evolution from water splitting photo-excitation of the photocatalyst generates electrons and holes in conduction band and valence band, respectively. Electrons reduce the protons to hydrogen, and holes oxidize water or other electron donor added. According to the band structure of PY-DHBD-COF and the redox potentials of water and ascorbic acid ($H_2A$), photocatalytic proton reduction into hydrogen and water oxidation into oxygen, as well as oxidation of the electron donor $H_2A$, are thermodynamically allowed[23].

**Photocatalytic hydrogen evolution.** After confirming the structures and fundamental photophysical properties of PY-DHBD-COF, the photocatalytic hydrogen evolution was investigated. The Pt co-catalyst was photodeposited in situ by reducing the $H_2PtCl_6$ precursor in a light reaction. Ascorbic acid was used as the sacrificial agent. Hydrogen was produced continuously under visible light irradiation (Fig. 4a). At a low loading of only 0.5 wt% Pt, the photocatalytic hydrogen evolution rate of PY-DHBD-COF reached 16,980 μmol g$^{-1}$ h$^{-1}$, which is a high level compared with previously reported values[16,22,24–28]. The photocatalytic hydrogen evolution increased with increasing Pt loading, and a top value of 71,160 μmol g$^{-1}$ h$^{-1}$ was achieved at 3 wt% Pt. The activity declined with a loading of 5 wt% Pt, which might be due

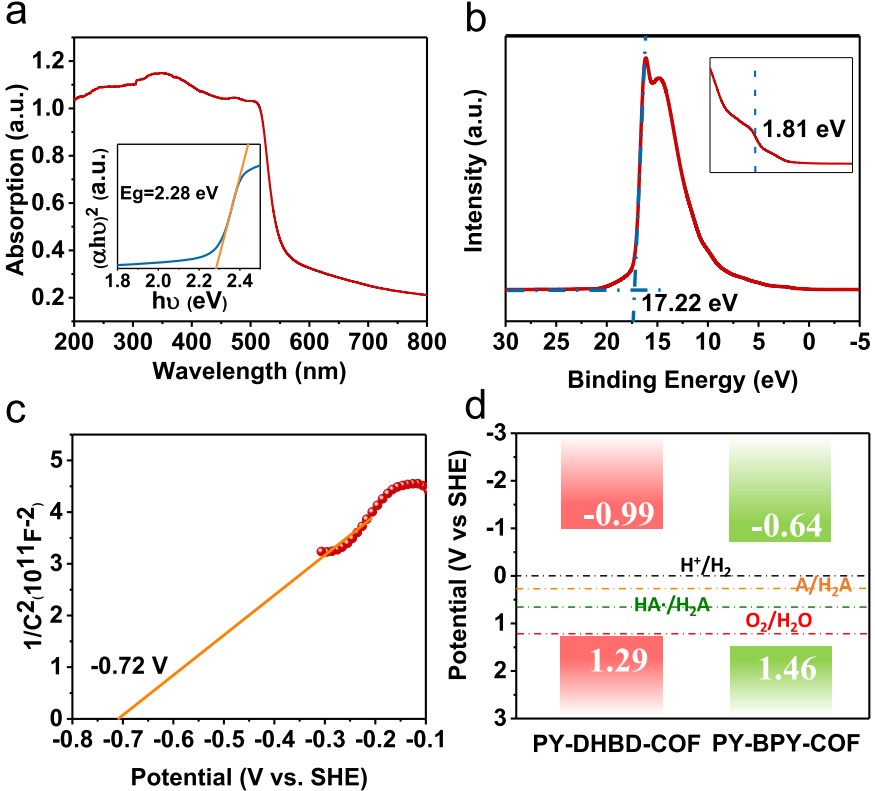

**Fig. 3 Optical and electronic properties of typical samples. a** UV-Vis diffuse reflectance spectrum of PY-DHBD-COF. Insert: Tauc plot in the form of $(\alpha h\upsilon)^2$ versus $h\upsilon$. **b** UPS spectrum of PY-DHBD-COF. **c** Mott–Schottky plots of PY-DHBD-COF. **d** Energy band positions of COFs.

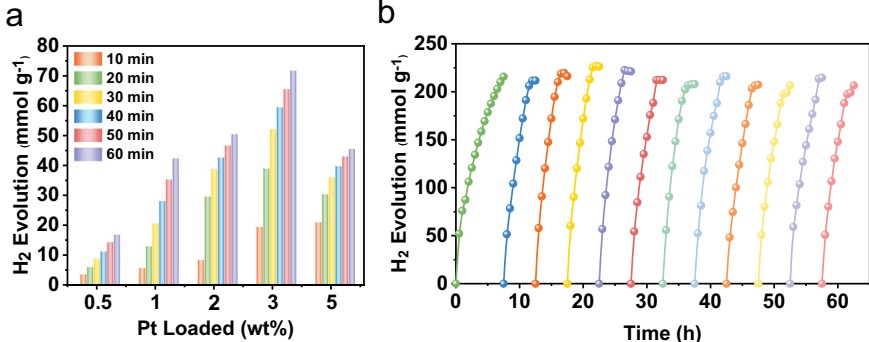

**Fig. 4 Hydrogen production performances. a** Time dependent hydrogen evolution for PY-DHBD-COF with different Pt loading amount. (10 mg catalyst was dispersed in 100 mL water, 10 mM ascorbic acid as electron donor, $H_2PtCl_6$ (0.376 g Pt $L^{-1}$) as Pt precursor, 300 W Xe lamp, λ > 420 nm). **b** Long-term hydrogen production for 3 wt% Pt loaded PY-DHBD-COF (Detailed data are listed in Supplementary Table 2).

to light shielding by the deposited Pt. In a long-term photo-catalytic experiment of 60 h with 3 wt% Pt-loaded PY-DHBD-COF, there was no significant decrease in the catalytic perfor-mance over time (Fig. 4b and Supplementary Table 2), suggesting the good reaction stability of PY-DHBD-COF. XRD and $N_2$ absorption-desorption isotherms of the 3 wt% Pt-loaded PY-DHBD-COF before and after the long-term photocatalytic reac-tion were conducted to check the structure stability. No obvious change was found in the XRD pattern (Supplementary Fig. 4), whereas the surface area and pore volume decreased based on the results of $N_2$ sorption experiment (Supplementary Fig. 5 and Supplementary Table 3), which indicates that there were losses in the crystallinity of the PY-DHBD-COF after the long-term pho-tocatalytic reaction. To further evaluate the performance of the photocatalyst, the apparent quantum yield (AQY), defined as the number of reacted electrons occurring per incident photon by the

system at a specified wavelength[29], was measured at 420 nm for 3 wt% Pt-PY-DHBD-COF. The AQY of 8.4% was determined, which is at a high level among the reported crystalline porous organic photocatalysts including COFs and MOFs (Supplemen-tary Table 4).

As a comparison, a similar COF (PY-BPY-COF) constructed with the same pyrene moiety and a different dipyridine moiety was investigated as a photocatalyst (Supplementary Fig. 6a). The basic structural characterizations of PY-BPY-COF, including FT-IR spectroscopy, PXRD, SEM, HR-TEM, and nitrogen adsorption isotherms, demonstrate that the crystalline COF was constructed successfully and that it had a framework and microporous structure similar to those of PY-DHBD-COF (Supplementary Figs. 6 and 7). The VBM and CBM potentials are confirmed to be thermodynamically feasible for water splitting according to the UV-Vis absorption, UPS, and Mott–Schottky tests (Fig. 3d and

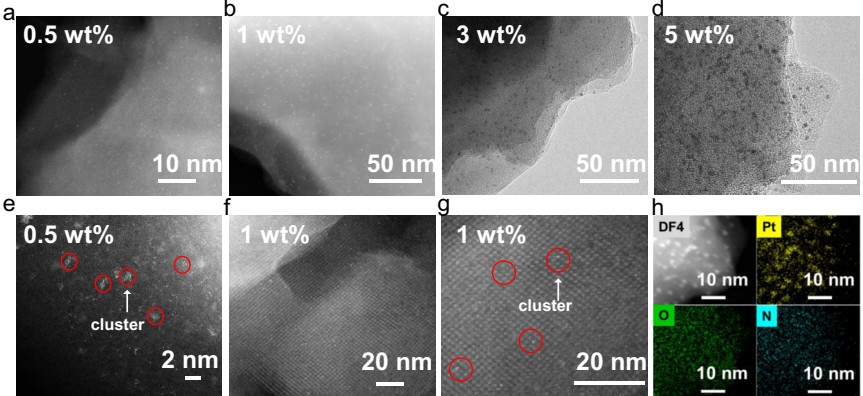

**Fig. 5 Morphologies of PY-DHBD-COF and platinum cocatalyst.** HR-TEM and HAADF-STEM images of PY-DHBD-COF loaded **a**, **e** 0.5 wt% Pt. **b**, **f**, **g** 1 wt% Pt. **c** 3 wt% Pt and **d** 5 wt% Pt. **h** EDX mapping images of 1 wt% Pt loaded PY-DHBD-COF.

Supplementary Fig. 8). In the photocatalytic reaction, the Pt-loaded PY-BPY-COF samples are able to continuously catalyse light-driven hydrogen generation (Supplementary Fig. 9). However, their HER activities are much lower than that of PY-DHBD-COF with the same Pt loading. The highest HER rate of PY-BPY-COF is 1204 μmol $g^{-1}$ $h^{-1}$ with a loading of 1 wt% Pt.

The reasons for the different HER activities of these two COFs were investigated to understand the important factors for efficient photocatalytic hydrogen evolution. First, the morphology of the loaded Pt was investigated by HR-TEM and HAADF-STEM. Regarding PY-BPY-COF, the loaded Pt particles are distributed randomly, and most are observed at the edge of the 2D COF layer in a large size (Supplementary Fig. 10). In contrast, for PY-DHBD-COF, the Pt species are uniformly and densely dispersed on the 2D layer of the COF material (Fig. 5a–d). At a loading of 0.5 wt% Pt, Pt exists in the form of both single atoms and small clusters (Fig. 5a, e). When the loading amount is increased to 1 wt%, more Pt clusters with a size of ~1 nm are observed, and the Pt species is further confirmed through EDX mapping (Fig. 5b, h). With a further increase in loading, the Pt particles gradually grow, and the size reaches ~3 nm at a loading of 3 wt% Pt and ~5 nm at a loading of 5 wt% Pt with the obvious lattice fringe of Pt (111) (Fig. 5c, d and Supplementary Fig. 11). The statistic diameter distributions showed that the sizes centred at a narrow range (Supplementary Fig. 12), further presenting the uniform deposition of Pt. These uniformly dispersed Pt co-catalysts can help easily trap localized photogenerated electrons on PY-DHBD-COF, thus suppressing charge combination during the long-distance transfer process. Moreover, the Pt clusters with small size exposed more active Pt atoms to catalyse the surface reaction, which would lead to high utilization efficiency of the noble metal. We normalized the HER activities according to the loading amount of Pt to assess the apparent utilization efficiency of the noble metal Pt co-catalyst (Supplementary Fig. 13). The activity order among different Pt loaded samples (1% > 0.5% > 2% > 3% > 5%) demonstrated the more efficient utilization of Pt at relative low loading amount with small Pt size.

Moreover, it is worth mentioning that the Pt clusters in 1 wt% Pt-PY-DHBD-COF are observed to deposit at a prior binding site beside the knot centre of the tetragonal porous framework (Fig. 5f, g). In view of the specific structure of PY-DHBD-COF, we speculate the binding site to be the area adjacent to the hydroxyl-O and imine-N on the framework. Thus small Pt clusters are possible to exist in the pore channels of PY-DHBD-COF. To check the entrance of Pt clusters into the pore channels, $N_2$ adsorption-desorption experiments were performed on 1 wt% Pt-PY-DHBD-COF, as well as a light-treated PY-DHBD-COF

obtained under the same photodepostion condition except for the existence of Pt precursor $H_2PtCl_6$ in the solution (Supplementary Fig. 14 and Supplementary Table 3). The porosity was retained on the light-treated PY-DHBD-COF, indicating the structural stability of PY-DHBD-COF under light irradiation during the photodeposition process. After Pt clusters were loaded, both surface area and pore volume decreased, which might be caused by the entrance of Pt clusters in the pore channels.

To reveal the original photodepostion behavior of Pt species on these two COFs, DFT calculations were performed. The B3LYP hybrid exchange-correlation functional[30–32] with the D3 version of Grimme's[33] dispersion implemented in the Gaussian 16 program[34] was used (for more details see the "Method" section). According to the COF structures, two simplified fragment models were built for PY-DHBD-COF and PY-BPY-COF (Supplementary Fig. 15a and b). In order to evaluate the effect of the –OH group, we also built the PY-BP-COF fragment as the comparison (Supplementary Fig. 15c). To study the initial process of photodeposition, adsorption of $PtCl_6^{2-}$ precursor on the COF fragments were simulated. Three possible adsorption sites have been considered for each fragment (Supplementary Fig. 16). The adsorption energy calculations of $PtCl_6^{2-}$ on PY-DHBD-COF, PY-BPY-COF and PY-BP-COF were conducted. The lowest adsorption energies of the $PtCl_6^{2-}$ ($\Delta E_{ads}$) on PY-DHBD-COF, PY-BPY-COF and PY-BP-COF are −8.96, −6.72, and −5.83 kcal/mol (Fig. 6a and Supplementary Table 5), respectively. The adsorption energy of $PtCl_6^{2-}$ on PY-DHBD-COF is 2.24 Kcal/mol higher than that on PY-BPY-COF and 3.13 Kcal/mol higher than that on PY-BP-COF. Therefore, the most stable adsorption site for $PtCl_6^{2-}$ on PY-DHBD-COF is around the –OH group and imine bond (Fig. 6a). Comparing the adsorption calculations of PY-DHBD-COF and PY-BP-COF, the introduction of the –OH group is favorable to stabilize the adsorbed $PtCl_6^{2-}$.

Furthermore, TD-DFT calculations were used to explore the electron transfer in the PY-DHBD-COF, PY-BPY-COF, and PY-BP-COF system, and the electron and hole distributions of the models were constructed by Multiwfn[35] and Visual Molecular Dynamics (VMD)[36]. The hole-electron analysis module of Multiwfn has been widely used to do the electron excitation analysis[37–40]. Based on the hole-electron theory, the S and D values are applied to evaluate the hole-electron separation, where S is the overlap integral of hole-electron distribution and D means the distance between centroid of hole and electron (the calculation details can be found in supporting information). The smaller S or larger D indicates the more evident hole-electron separation. The PY-BPY-COF show the lowest S (Supplementary

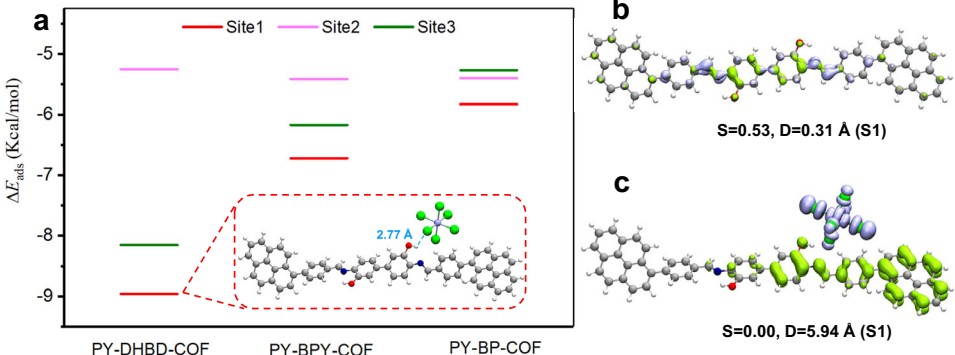

**Fig. 6 Theorectical understanding of the photodeposition. a** The adsorption energy of $PtCl_6^{2-}$ on three sites of PY-DHBD-COF, PY-BPY-COF, and PY-BP-COF (Insert: the most stable adsorption configuration of PY-DHBD-COF-$PtCl_6^{2-}$). The hole (lime) and electron (violet) distribution of S1 excited state on PY-DHBD-COF (**b**) and the complex PY-DHBD-COF-$PtCl_6^{2-}$ (**c**). S is the overlap integral of hole-electron distribution and D means the distance between centroid of hole and electron. The balls in different colors represent different atoms: H, white; C, dark gray; N, blue; O, red.

Figs. 17–19, the S is 0.27, the D is 0.0 because of the centrosymmetric structure), which denotes the best hole-electron separation. Nevertheless, the weak binding ability of $PtCl_6^{2-}$ precursor may cause the non-uniform deposition of Pt. The PY-DHBD-COF and PY-BP-COF has similar S of 0.53, while PY-DHBD-COF shows significant larger D, indicating that electron-hole separation on PY-DHBD-COF is more efficient. The introduction of –OH group may promotes the separation of electrons and holes comparing PY-DHBD-COF and PY-BP-COF. The hole-electron distribution analysis (Fig. 6b) and the odd electron density (OED) analysis (Supplementary Fig. 17e) of PY-DHBD-COF show that the photogenerated electrons mainly locate around the imine-N sites. Combining with the most stable adsorption configuration of PY-DHBD-COF-$PtCl_6^{2-}$ (Fig. 6a), electron transfer to the imine-N sites is benefit to the photoreduction of Pt because the Pt species tend to adsorbed around this site. Subsequently, the electron transfer in the most stable PY-DHBD-COF-$PtCl_6^{2-}$ configuration was investigated. The hole and electron distribution analysis shows that holes almost distribute on the PY-DHBD-COF fragment entirely, while electrons are on $PtCl_6^{2-}$ (Fig. 6c). The OED analysis further confirm that the odd electron mainly distribute on Pt atom (Supplementary Fig. 17f). These results illustrate one thing that when PY-DHBD-COF is excited, the electrons mainly transfer to $PtCl_6^{2-}$ and caused the photodeposition process. Furthermore, we optimized the geometry of the PY-DHBD-COF-$PtCl_6^{2-}$ in its singlet excited state S1 using TD-DFT. The full optimized configuration shows that two Pt-Cl bonds are significantly elongated and a weak hydrogen bond is formed between one Cl atom and H on –OH group (Supplementary Fig. 20). This loosed configuration is favorable to loss $Cl^-$ and form a $Pt(III)Cl_5^{2-}$ species and lead the following photodeposition reaction. Two possible photodeposition reaction paths are calculated as shown in Supplementary Fig. 21.

Combining with the gradually changing morphology observed by TEM and the DFT/TD-DFT calculations, we propose that Pt atoms are likely to be first adsorbed on the hydroxyl and imine site on the structural unit of PY-DHBD-COF and further reduced into clusters and metallic particles with increasing precursor concentration during the photodeposition process. XPS spectra of Pt 4f show that Pt species exist in the oxidation state of $Pt^{II}$ in 0.5 wt% Pt-PY-DHBD-COF and the metallic state of $Pt^0$ with an increase in loading (Supplementary Fig. 22), further confirming the gradual deposition process. Due to the uniformly dispersed binding site for Pt species, the Pt co-catalyst can spread all over the 2D layer, thus greatly boosting the electron transfer from the COF to the Pt co-catalyst.

Furthermore, electron transfer under light irradiation was investigated through steady-state and time-resolved fluorescence spectroscopy. Fluorescence quenching are observed for both PY-DHBD-COF and PY-BPY-COF after loading 1 wt% Pt (Fig. 7a, b), indicating electron transfer from the COF substrate to the Pt co-catalyst. Compared with the 80% fluorescence quenching of Pt-PY-BPY-COF, the nearly complete fluorescence quenching of Pt-PY-DHBD-COF reveals more efficient electron transfer. In the time-resolved fluorescence spectrum test, the fluorescence lifetimes of Pt-loaded COFs are lower than that of bare COF (Fig. 7c and d), indicating suppressed $e- -h+$ recombination[41]. As a result, the highly dispersed Pt particles on PY-DHBD-COF promote efficient electron transfer to stop e-h recombination, thereby enhancing the HER.

It is well known that metallic Pt is able to collect photogenerated electrons and catalyse proton reduction on the surface owing to its low work function and proper adsorption-desorption energy of $H^+$ and $H_2$. In this work, due to the existence of binding and reducing sites for $PtCl_6^{2-}$ in each periodic unit PY-DHBD-COF, uniformly and densely dispersed metallic Pt clusters were deposited on the COF, which not only facilitated the fast and unique separation of electrons and holes but also provided a high atom-utilization efficiency for hydrogen evolution, therefore leading to the excellent HER performance. This suggests the superiority and availability of the site binding strategy to deposit Pt clusters as co-catalysts. Although the clusters grew into larger particles when increasing the precursor concentration, the high dispersion of Pt particles on the COF layer still improved the local charge separation and resulted in a high activity of 71,160 $\mu mol\ g^{-1}\ h^{-1}$. It is noted that Pt clusters have not occupied all the chelating sites yet. To increase the loading of Pt clusters for more efficient hydrogen evolution, the controlled growth of Pt clusters is still being studied in our lab.

## Discussion

In summary, we reported the designed synthesis of a crystalline 2D covalent organic framework PY-DHBD-COF bearing a binding site for the in situ photodeposition of Pt clusters. The regularly distributed hydroxyl group and adjacent imine-N on the structural units contribute to optimal adsorption sites for the Pt precursor, leading to the uniform in situ photodeposition of Pt co-catalysts on the scale from single atoms to small clusters and nanoparticles, which greatly improved the spatial charge transfer between the COF and co-catalyst. By exposing a more catalytically active surface, the Pt cluster-loaded COF showed an efficient utilization of Pt. This work provides a strategy for the

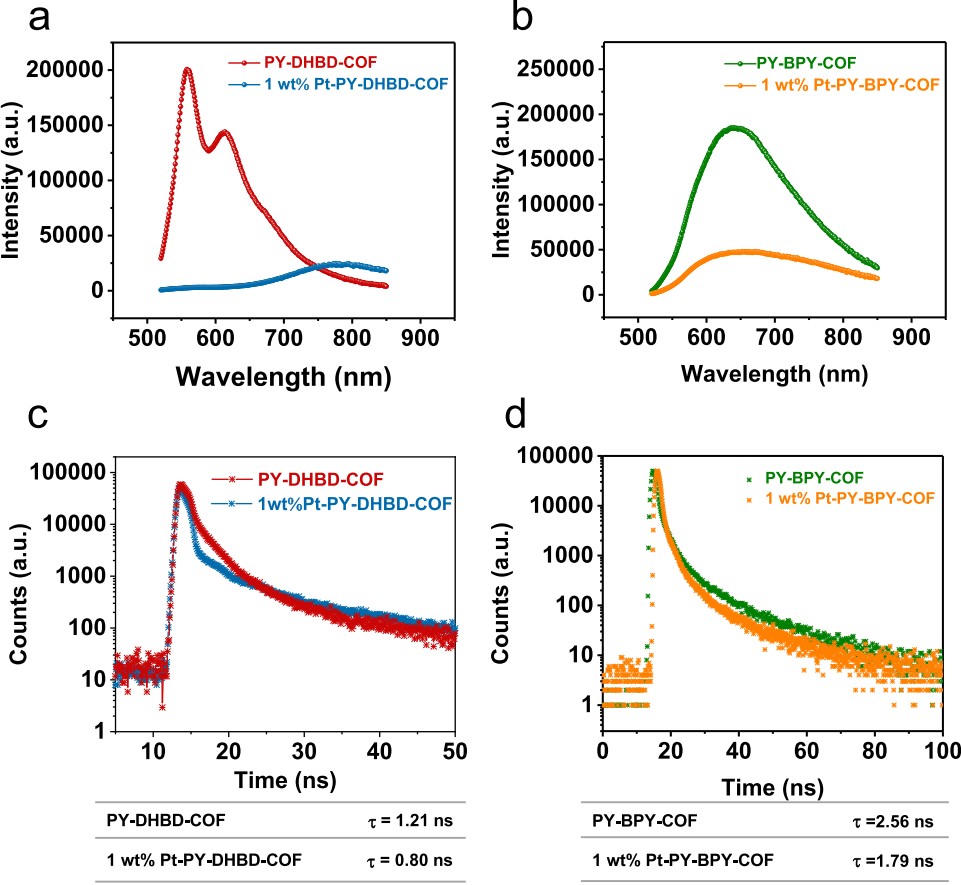

**Fig. 7 Fluorescence emission and lifetime of the typical samples. a, b** Fluorescence spectra and **c, d** fluorescence lifetime decay of PY-DHBD-COF, PY-BPY-COF and the corresponding Pt (1 wt%) loaded samples. PY-DHBD-COF and PY-BPY-COF were excited with a $\lambda_{ex} = 374$ nm laser and the emissions were measured at 560 nm and 620 nm, respectively.

establishment of a micro-environment for co-catalyst deposition on a photocatalyst for more efficient photocatalysis.

## Methods

**Synthesis of PY-DHBD-COF.** An n-butyl alcohol (n-BuOH)/o-dichlorobenzene (o-DCB)/6 M AcOH (5/5/1 by vol., 1.1 mL) mixture of 1,4-dihydroxybenzidine (0.04 mmol, 8.6 mg) and 1,3,6,8-tetra(4-formylphenyl)pyrene (0.02 mmol, 12.3 mg) in a Pyrex tube (10 mL) was degassed by three freeze-pump-thaw cycles. The tube was sealed off and heated at 120 °C for 3 days. The precipitate was collected by centrifugation and washed with anhydrous THF 5 times and acetone twice. The powder was dried at 120 °C under vacuum overnight to give PY-DHBD-COF in an isolated yield of 85%.

**Synthesis of PY-BPY-COF.** An n-butyl alcohol (n-BuOH)/o-dichlorobenzene (o-DCB)/6 M AcOH (5/5/1 by vol., 1.1 mL) mixture of 2,2'-bipyridine-5,5'-dicarbaldehyde (0.04 mmol, 8.5 mg) and 4,4',4'',4'''-(pyrene-1,3,6,8-tetrayl) tetraaniline (0.02 mmol, 11.3 mg) in a Pyrex tube (10 mL) was degassed by three freeze-pump-thaw cycles. The tube was sealed off and heated at 120 °C for 3 days. The precipitate was collected by centrifugation and washed with anhydrous THF 5 times and acetone twice. The powder was dried at 120 °C under vacuum overnight to produce PY-BPY-COF in an isolated yield of 82%.

**Characterizations.** The Brunauer–Emmett–Teller (BET) surface areas of COFs were measured at 77 K by using a Quantachrome Automated Surface Area & Pore Size Analyzer. Pore size distributions were estimated by nonlocal density functional theory (NLDFT). The powder X-ray diffraction (PXRD) pattern was recorded with a Cu-Kα X-ray radiation source ($\lambda = 0.154056$ nm) by a Rigaku MiniFlEX 600 instrument over the range of $2\theta = 2.0$–40.0° with a step size of 0.02° per step. The FT-IR spectra were recorded by a Thermo Nicolet iS50 instrument in the range from 400 to 4000 cm$^{-1}$. Solid-state $^{13}$C CP/MAS NMR spectra were recorded on a 400WB S2 AVANCE III instrument (Bruker, Switzerland) along with a 400 MHz spectrophotometer at 298 K. Morphological information for the COFs was obtained from field-emission scanning electron microscopy (SEM, FEI Nano 450), transmission electron microscopy (TEM, JEOL JEM 2100 f) and high-angle

annular dark-field scanning transmission election microscopy (HAADF-STEM, FEI Themis Z). X-ray photoelectron spectroscopy (XPS) was conducted with an ESCALAB 250 instrument with a monochromatic Al Kα X-ray source. Thermo-gravimetric analyses (TGA) were recorded with a Netzsch Model STA 449 C micro-analyser heated from 25 °C to 900 °C in a nitrogen atmosphere. UV-visible absorption spectra of the polymers were measured with a Shimadzu UV-2550 UV-vis spectrometer by measuring the reflectance of powders in the solid-state. High-resolution valence band ultraviolet photoelectron spectra (UPS) were obtained from a Thermo Fisher Scientific Escalab 250Xi instrument. Fluorescence spectra and fluorescence lifetimes were measured using a FLS1000 Edinburgh Instruments spectrofluorometer.

**Photocatalytic hydrogen evolution measurement.** Photocatalytic hydrogen evolution measurements were carried out in a Pyrex top irradiation reaction vessel connected to a glass closed gas circulation system (Labsolar 6 A, Perfect Light). For each reaction, 10 mg of photocatalyst was dispersed in 100 mL of $H_2O$, 10 mM ascorbic acid was used as the sacrificial agent, and the mixture was ultrasonicated for 30 min to obtain a uniform dispersion before adding an appropriate amount of $H_2PtCl_6$ aqueous solution (0.376 g Pt/L) as the Pt precursor. The above suspension was degassed and kept at 20 °C by using circulated cooling water. A xenon lamp equipped with a long-pass filter (>420 nm) was used as the light source. The gas products were analyzed by an online 8890 GC System (Agilent) with Ar as the carrier gas. After the photocatalysis experiment, the photocatalysts were collected by washing with water and acetone and then dried in a 120 °C vacuum oven. For the long-term photocatalytic experiment, 2 mM ascorbic acid was added initially, and the system was degassed and 2 mM ascorbic acid was replenished when the amount of hydrogen no longer increased.

**Electrochemical measurements.** A Mott–Schottky plot was recorded on a NOVA II electrochemical workstation with a standard three-electrode system, containing PY-DHBD-COF-coated FTO as the working electrode, a Pt plate as the counter electrode and a Ag/AgCl electrode as the reference electrode. A 0.2 M $Na_2SO_4$ solution (pH = 2.5) was used as the electrolyte. The applied potentials vs. Ag/AgCl

were converted to RHE potentials using the following equation:

$$E_{\mathrm{RHE}} = E_{\mathrm{Ag/AgCl}} + 0.0591\,\mathrm{pH} + E^{\theta}_{\mathrm{Ag/AgCl}}(E^{\theta}_{\mathrm{Ag/AgCl}} = 0.199\,\mathrm{V}) \tag{1}$$

**DFT calculations**. The photo-excitation of COFs was evaluated by using a combination of DFT for the ground states and TD-DFT for the excited state. We use a representative fragment rather than the periodic COF framework, as the description of the charged system is difficult for the periodic system[42]. The DFT calculations were carried out using the B3LYP hybrid Exchange-Correlation (XC) functional[30–32] with the D3[33] version of Grimme's dispersion as implemented in Gaussian 16[34]. The ground-state geometry optimization calculation employed the 6-31G(d) basis set for C, H, O, N, Cl, and the Stuttgart/Dresden (SDD)[43] basis set with the corresponding effective core potential (ECP) was employed for Pt[44]. Thus, we still use SDD with corresponding ECP for Pt in the single point energy calculation. Solvent effects of water were approximated by the PCM solvation model[45]. PCM solvent effects were considered both in single point calculations and full geometry optimizations including in the case of TD-DFT calculations.

The TD-DFT calculations were performed with PBE0-D3 method[46] instead of B3LYP, as the PBE0 functional is more reliable for the excitation energies calculation[16]. The adsorption Gibbs free energy ($\Delta G_{\mathrm{ads}}$) and binding energy ($\Delta E_{\mathrm{ads}}$) are calculated with the following equations:

$$\Delta G_{\mathrm{ads}} = G_{\mathrm{COF-PtCl_6^{2-}}} - G_{\mathrm{COF}} - G_{\mathrm{PtCl_6^{2-}}} \tag{2}$$

$$\Delta E_{\mathrm{ads}} = E_{\mathrm{COF-PtCl_6^{2-}}} - E_{\mathrm{COF}} - E_{\mathrm{PtCl_6^{2-}}} \tag{3}$$

## Data availability

All data supporting the findings of this study are available from the source data. The relevant DFT optimised structures are provided in the Supplementary Data. Source data are provided with this paper.

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

## Acknowledgements

This work was supported by the National Key Research and Development Program of China (No. 2017YFA0204800), National Natural Science Foundation of China (No. 22002070) and Natural Science Foundation of Shandong Province (No. ZR2020QB059) and the Fundamental Research Funds of Shangdong University (No. 2019GN108). Z.L. thank the financial support from the Program of Young Scholars Future Program of Shandong University.

## Author contributions

Z.L. and W.Q.D. designed the whole project. Y.M.L. executed the catalysts syntheses and the catalytic activity evaluation. L.Y. performed the DFT calculations. L.S., H.L.W., and Y.L.Z. offered help in theoretical calculation. H.J.H., X.F., D.Y.Z., and Y.Q. offered help in adsorption characterization, quantum efficiency measurement, spectroscopic and electrochemical characterizations. All authors were involved in the writing of the paper.

## Competing interests

The authors declare no competing interests.
