## [Peer Review File · Nature Communications]

In situ photodeposition of single platinum clusters on a covalent organic framework for photocatalytic hydrogen productionREVIEWER COMMENTS

Reviewer #1 (Remarks to the Author):

In this manuscript, Deng and coworkers reported a strategy to in situ photodeposit uniform platinum clusters on a covalent organic framework by introducing adjacent hydroxyl-O and imine-N as chelating sites of platinum atoms. It is a really interesting and important result which shows a new typical direction for precise controlling of cocatalyst deposition to significantly improve the efficiency of photocatalyst by utilizing the structural diversity and designability of COF. This strategy makes the materials form an excellent composite with enhanced electron transfer and separation ability and thereby presenting extraordinary performance in light driven hydrogen production. Therefore, I would recommend its publication after addressing several minor issues.

1. It is suggested to add the diameter distributions of Pt clusters to show the uniform deposition of Pt.
2. The authors have normalized the hydrogen evolution rate according to the loading amount of Pt with a measuring unit of " $\mu\text{mol g}^{-1} \text{h}^{-1} (1 \text{ wt\% Pt})^{-1}$ ". This is unnormal and rarely used to assess the activity. It is recommended to give a normalized value by only Pt or only catalyst.
3. It is recommended to add a description of the principle of light driven hydrogen production.

Reviewer #2 (Remarks to the Author):

Having read the manuscript by Li and co-workers I believe the results presented are interesting but that there are minor issues with the experimental data and their analysis and very major issues with the computational data and their analysis. Specifically:

The supporting information contains what appears to be a structure of the COF obtained from the powder diffraction x-ray data, but I don't seem to be able to find any data about the refinement of the x-ray data that resulted in that structure. I also note that the authors only seem to consider AA and AB stackings but that for other COFs slightly offset AA stackings, AA' stackings if you like, were found to best fit the x-ray patterns measured.

The authors compare the hydrogen evolution rate of their COF with those reported for other COFs in the literature. However, those values were measured at other setups, and one should be careful with a straight comparison of hydrogen evolution rates measured at different setups as they depend strongly on the light flux encountered by the catalyst, which itself depends on the lamp spectrum, reactor

thickness etc. etc. A comparison in terms of external quantum efficiency should be fairer as this should be less sensitive to the setup used. Taking the value reported in the manuscript, 8.4% at 420 nm, the performance is similar to that of other COFs in the literature. See page 1183 of the Cooper Sulfone COF paper for an overview of COF quantum efficiencies.

The authors might want to include a larger version of Fig. 3B in the supporting information as it's hard to read of. Similarly, they might want to put a table containing the underlying data in the supporting information.

I don't see the justification for the normalisation of the HER with respect to 1 wt% Pt in Fig. 3C obtained apparently by dividing the HER by the platinum loading. I think it's better for the authors to present the unadulterated data in Fig. 3C. Also, because this HER data has been measured on different setups I am, as discussed above, sceptical about the meaning of such a comparison.

DFT calculations of the adsorption of a species on another species really should employ an empirical dispersion correction, e.g. the Grimme D3 dispersion correction giving B3LYP+D3 in the case of B3LYP, which does not seem to be the case here. Please note, that because of the presence of Pt use of Grimme's D4 dispersion correction is probably preferred over the D3 correction but I don't know if the former has already been implemented for Gaussian.

I might be missing something but what happened to the hydroxyl protons in Fig. 5A and S10C, E and G? Are the authors somehow assuming that the hydroxyls are deprotonated under the conditions where Pt is deposited? This feels rather odd and doesn't seem to be discussed in the manuscript. Overall, it makes me uncertain about the accuracy and significance of the values of the binding energies reported.

The authors write "For the geometry optimizations and frequency calculations, we used the standard 6-31g* basis set, while single-point energy calculations were performed at each stationary point using the extended 6-31++g**" in the methods section. However, I am not aware that there is a 6-31g* or 6-31++g** basis-set for Pt, and even if there was the high-Z of Pt would probably require a relativistic ECP. Again, this makes me uncertain about the accuracy and significance of the values of the binding energies reported.

In the supporting information adsorption values for I assume, based on Fig. S10A and S10B, Pt(IV)Cl₄ and Pt(II)Cl₂ are reported, with the values quoted in the main text apparently those predicted for the case of Pt(II)Cl₂. Why were these specific platinum complexes studied? I note that experimentally the Pt source is H₂Pt(IV)Cl₆, which I assume is in water present as [Pt(IV)Cl₆]²⁻, which appears far removed from Pt(II)Cl₂. I can't find any discussion justifying the choice of Pt cluster in the calculations, as a matter of fact I don't think the main text mentions the type of Pt cluster used in the calculations at all. There is

also the issue that the Pt deposition chemistry takes place in water and not in the gas-phase and that really an implicit solvation model should have been used, possibly in combination with explicit water to complete the coordination sphere of platinum. Again, this makes me uncertain about the accuracy and significance of the values of the binding energies reported.

Overall, I don't think the calculated Pt complex adsorption energies can be trusted as it stands and I remain sceptical that even if they were correct that they would necessarily support the mechanism of Pt deposition proposed by the authors over alternative deposition pathways or why Pt deposition is different for this COF relative to other COFs in the literature.

Why did the authors switch from B3LYP to HSEH1PBE for obtaining the molecular orbitals? Again this does not seem to be discussed in the manuscript.

The authors should include all relevant DFT optimised structures in the supporting information in the form of machine-readable files, i.e. as e.g. xyz files rather than coordinated pasted in a PDF. Similarly, they should really include a table with the total energies, entropies and free energies of all species in the supporting information.

Reviewer #3 (Remarks to the Author):

In this work titled "In situ photodeposition of platinum clusters on a covalent organic framework for light driven hydrogen production", the authors report the preparation of photocatalytic systems based on covalent organic frameworks and photodeposited platinum clusters as cocatalysts for the hydrogen evolution reaction. Emphasis is given to the successful photodeposition of well-dispersed platinum clusters on the PY-DHBD-COF photocatalyst containing hydroxyl and imine functional groups, which resulted in one of the highest hydrogen evolution rates for this type of materials. I find this research study very interesting and I thus recommend this manuscript for publication in Nature Communications after the following issues are addressed:

1) The authors compare MOFs and COFs as photocatalysts, stating that the former class offers very limited stability as opposed to COFs and their application in photocatalysis is limited. However, both statements are not necessarily true as there is a large number of MOFs that are very stable photocatalysts (e.g., MIL-125, UIO-66, MIL-167, Al-PMOF to name a few, Energy Environ. Sci., 2015, 8, 364, ACS Appl. Mater. Interfaces 2021, 13, 12, 14239–14247), while COFs can suffer from stability especially under elevated temperature. I thus believe that this part of the introduction should be adjusted so as to highlight the real advantage of COF photocatalysts.

2) The authors state: ‘.. compared with traditional molecular catalysts, COFs lock the photoactive building units in a rigid structure to prevent photocorrosion and improve the life of the excited state’. Please provide references of research studies in the literature supporting this argument.

3) The in-plane pore channels of the proposed AA-stacked PY-DHBD-COF demonstrate a size of around 2.5 nm. Such value could allow the presence of Pt clusters within these pore channels, which the authors do not sufficiently describe. I thus recommend a description of the potential location of the Pt clusters. Carrying out N₂ adsorption-desorption experiments for the COFs with photodeposited Pt clusters would show how much the porosity is compromised due to the presence of Pt clusters within the pores.

4) After the photocatalytic recycling experiments the preservation of the crystallinity was evaluated through PXRD. However, PXRD detects only significant changes in crystallinity (Angew. Chem. 2019, 131, 18007 – 18012). To ensure that the crystallinity and porosity of the COF is retained after the experiments, please provide N₂ sorption isotherms of the photocatalyst after the long-term photocatalytic test.

5) Metrics based on the incident or absorbed radiation (e.g., Apparent quantum yield/ quantum efficiency) are more appropriate to evaluate the photocatalytic activity and to facilitate the comparison with other photocatalysts using different set-up. Please provide the Apparent Quantum Yield (at wavelength >420 nm) or Quantum efficiency of the best photocatalyst.

6) The authors state: ‘.. the fluorescence lifetimes of the PY-DHBD-COF samples are shorter than those of the corresponding PY-BPY-COF samples (1.09 ns < 1.79 ns; 1.93 ns < 2.56 ns), indicating more efficient charge separation of the PY-DHBD-COF samples.’ However, shorter lifetimes correspond to faster recombination. What do the authors mean by this statement?

7) At the transient photoluminescent experiments (TPL), the recorded PL emission was at 620 nm, which corresponds to the highest PL intensity of PY-BPY-COF, but not to that of PY-DHBD-COF. Therefore, I believe that the TPL of PY-DHBD-COF at its maximum PL intensity (around 570 nm) should be provided for a valid comparison.

Point-by-point response to the reviewers' comments on NCOMMS-21-21627

Reviewer #1: In this manuscript, Deng and coworkers reported a strategy to in situ photodeposit uniform platinum clusters on a covalent organic framework by introducing adjacent hydroxyl-O and imine-N as chelating sites of platinum atoms. It is a really interesting and important result which shows a new typical direction for precise controlling of cocatalyst deposition to significantly improve the efficiency of photocatalyst by utilizing the structural diversity and designability of COF. This strategy makes the materials form an excellent composite with enhanced electron transfer and separation ability and thereby presenting extraordinary performance in light driven hydrogen production. Therefore, I would recommend its publication after addressing several minor issues.

We are grateful to the reviewer's positive comments on our work. We have carried out supplementary experiments and made modifications according to the constructive suggestions. Detailed responses are provided as follows.

Comments 1:

It is suggested to add the diameter distributions of Pt clusters to show the uniform deposition of Pt.

Reply:

According to reviewer's comments, we measured the diameter distributions of Pt on PY-DHBD-COF with different Pt loads, and the results were shown in Figure R1. For low loads of 0.5% and 1%, the size centered at around 1.2 nm, and for load of 3%, the size centered at around 3 nm, and for high load of 5%, the size distribution became a little broad and the diameter increased to 5 nm. We added Figure R1 into the revised electronic supplementary information as Figure S12, and added the sentence "The statistic diameter distributions showed that the sizes centred at a narrow range (Figure S12), further presenting the uniform deposition of Pt" in the revised manuscript.

Figure R1 (Figure S12 in Supporting information). The diameter distributions of Pt on PY-DHBD-COF with different Pt loads

Comments 2:

The authors have normalized the hydrogen evolution rate according to the loading amount of Pt with a measuring unit of “ $\mu\text{mol g}^{-1} \text{h}^{-1} (1 \text{ wt}\% \text{ Pt})^{-1}$ ”. This is unnormal and rarely used to assess the activity. It is recommended to give a normalized value by only Pt or only catalyst.

Reply:

We presented the measuring unit of “ $\mu\text{mol g}^{-1} \text{h}^{-1} (1 \text{ wt}\% \text{ Pt})^{-1}$ ” for evaluating the utilization efficiency of noble metal Pt for hydrogen evolution. According to the reviewer’s suggestion, we normalized the hydrogen evolution only by Pt, and the results was shown in figure R2, which revealed that the utilization efficiency was the highest at low Pt loadings to form Pt cluster with small size to expose more Pt active Pt atoms. What’s more, the direct comparison between our results and those in other reported works was canceled as the experiments were conducted under different setups. Finally, We added Figure R2 in the

revised electronic supplementary information as Figure S13, and added the sentence “Moreover, the Pt clusters with small size exposed more active Pt atoms to catalyze the reaction, which would lead to high utilization efficiency of the noble metal. We normalized the HER activities according to the loading amount of Pt to assess the apparent utilization efficiency of the novel metal Pt co-catalyst (Figure S13). The activity order among different Pt loaded samples (1%>0.5%>2%>3%>5%) demonstrated the more efficient utilization of Pt at relative low loading amount with small Pt size.” in the revised manuscript.

Figure R2 (Figure S13 in Supporting information). HER activity normalized according to the loading amount of Pt.

Comments 3:

It is recommended to add a description of the principle of light driven hydrogen production.

Reply:

According to reviewer’s comments, we added the sentence “As we all know that in photocatalytic hydrogen evolution from water splitting photoexcitation of the photocatalyst generates electrons and holes in conduction band and valence band, respectively. Electrons reduce the protons to hydrogen, and holes oxidize water or other electron donor added. According to the band structure of PY-DHBD-COF and the redox potentials of

water and ascorbic acid (H_2A), photocatalytic proton reduction into hydrogen and water oxidation into oxygen, as well as oxidation of the electron donor H_2A , are thermodynamically allowed" in the revised manuscript.

Reviewer #2: Having read the manuscript by Li and co-workers I believe the results presented are interesting but that there are minor issues with the experimental data and their analysis and very major issues with the computational data and their analysis.

We are grateful to the reviewer's valuable suggestions on our work. According to the reviewer's suggestions, we have recalculated all the DFT calculations and revised our manuscript carefully. Detailed responses are provided as follows.

Comments 1:

The supporting information contains what appears to be a structure of the COF obtained from the powder diffraction x-ray data, but I don't seem to be able to find any data about the refinement of the x-ray data that resulted in that structure. I also note that the authors only seem to consider AA and AB stackings but that for other COFs slightly offset AA stackings, AA' stackings if you like, were found to best fit the x-ray patterns measured.

According to the reviewer's comment, Pawley refinement of the XRD data was conducted and the results were shown in Figure.R3. Pawley refinement confirmed that the diffraction pattern of PY-DHBD-COF was consistent with an orthorhombic lattice with unit cell parameters ($a = 39.355(3) \text{ \AA}$, $b = 41.382(1) \text{ \AA}$, $c = 3.894(7) \text{ \AA}$) similar to an idealized AA-stacking model. Therefore, we propose that PY-DHBD-COF have AA layer stackings, rather than AB stacking. In addition, the simulated XRD patterns of a variety of possible offset AA stackings, including AA flipped, AA slipped-1 (with offset of 1.44 \AA in the (110) direction) and AA slipped-2 (with offset of 2.88 \AA in the (110) direction), were compared with the experimental XRD data of PY-DHBD-COF (Figure R4), and the XRD data do not allow us to distinguish between these. For the purposes of structural comparisons with experimental data, we refer to the idealized, perfectly eclipsed AA stacking patterns, because properties such as porosity are not greatly affected by small shifts in the relative orientation of the layers. Similarly, the possible structure of PY-BPY-COF was predicted by the XRD refinement and the comparison of simulated XRD patterns in different stacking modes (Figure R5 and R6), and we propose that PY-BPY-COF have AA layer stackings with a orthorhombic lattice.

Finally, we added Fig.R3 into the revised manuscript as Figure 1d, and added Fig.R4, R5 and R6 into the revised supplementary information as Figure S1c, S6, respectively. For the clear display of figure 1d, we moved the solid state ^{13}C -NMR spectrum and SEM image

of PY-DHBD-COF into support information. The description sentences of “Pawley refinements show that the diffraction pattern of PY-DHBD-COF was consistent with an orthorhombic lattice($R_p = 5.58\%$, $R_{wp} = 7.74\%$) with unit cell parameters ($a = 39.355(3)$ Å, $b = 41.382(1)$ Å, $c = 3.894(7)$ Å), which matches well with the idealized AA-stacking model. Moreover, a variety of other possible AA stackings, including AA flipped and slipped models, were considered, and the simulated XRD patterns are almost same (Figure S1c), making it difficult to distinguish between these. Therefore, we refer to the idealized, perfectly eclipsed AA stacking model due to its lowest energy (Table S1)” was added in the revised manuscript.

Figure R3 (Figure 1d in Manuscript). Experimental and simulated PXRD patterns of PY-DHBD-COF.

Figure R4 (Figure S1c in Supporting information). Comparison of experimental and simulated powder X-ray diffraction patterns of several possible stacking models of PY-DHBD-COF.

Figure R5 (Figure S6c in Supporting information). Experimental and simulated PXRD patterns of PY-BPY-COF.

Figure R6 (Figure S6d in Supporting information). Comparison of experimental and simulated powder X-ray diffraction patterns of several possible stacking models of PY-BPY-COF.

Comments 2:

The authors compare the hydrogen evolution rate of their COF with those reported for other COFs in the literature. However, those values were measured at other setups, and one should be careful with a straight comparison of hydrogen evolution rates measured at different setups as they depend strongly on the light flux encountered by the catalyst, which itself depends on the lamp spectrum, reactor thickness etc. etc. A comparison in terms of external quantum efficiency should be fairer as this should be less sensitive to the setup used. Taking the value reported in the manuscript, 8.4% at 420 nm, the performance is similar to that of other COFs in the literature. See page 1183 of the Cooper Sulfone COF paper for an overview of COF quantum efficiencies.

Reply:

Thanks for the reasonable suggestion. Accordingly, we compared the external quantum efficiency (EQE) of some representative crystalline porous organic photocatalysts, including COFs and MOFs, reported in literature with that of PY-DHBD-COF as listed in Table R1, so as to evaluate the catalytic performance more fairly. The EQE of 8.4% at 420 nm in our work is similar to a high value of 8.45% reported by L. Chen et al. (Angew. Chem. Int. Ed., 2020, 59, 16902-16909). Finally, we deleted the straight comparison of hydrogen evolution rates in Figure 3c, and added Table R1 in the revised supplementary information as Table S4. The sentence “An external quantum efficiency (EQE) of 8.4% was determined for 3 wt% Pt-PY-DHBD-COF at 420 nm, which is at a high level among the reported crystalline porous organic photocatalysts including COFs and MOFs (Table S4)” was added in the revised manuscript.

Table R1 (Table S4 in Supporting information). The photocatalytic hydrogen evolution performance comparison of PY-DHBD-COF with other representative COF and MOF based photocatalysts.

Entry	Co-catalyst	Sacrificial reagent	Light Source	HER ($\mu\text{mol g}^{-1} \text{h}^{-1}$)	EQE (at 420 nm) (%)	Ref.
PY-DHBD-COF	Pt 0.5wt%	Ascorbic acid	> 420 nm	16980	--	This work
PY-DHBD-COF	Pt 1wt%	Ascorbic acid	> 420 nm	42432	--	This work
PY-DHBD-COF	Pt 2wt%	Ascorbic acid	> 420 nm	56712	--	This work
PY-DHBD-COF	Pt 3wt%	Ascorbic acid	> 420 nm	71160	8.4	This work
PY-DHBD-COF	Pt 5wt%	Ascorbic acid	> 420 nm	48912	--	This work
Tp-2C/BPy ²⁺ -COF (19.10%)	Pt 3wt%	Ascorbic acid	> 420 nm	34600	6.93	1 ³
Py-CITP-BT-COF	Pt 5wt%	Ascorbic acid	> 420 nm	8875	8.45	2 ⁴

FS-COF	Pt 8wt%	Ascorbic acid	> 420 nm	10100	3.2	3 ⁵
g-C ₁₈ N ₃ -COF	Pt 3wt%	Ascorbic acid	> 420 nm	2920	4.84	4 ⁶
MIL-125/Au	Pt 0.49wt%	TEOA	> 420 nm	1743	--	5 ⁷
Uio-66-NH ₂	Pt 0.65wt%	Ascorbic acid	> 420 nm	1528	2.3	6 ⁸
MIL-125-NH ₂	Pt 0.45wt%	TEOA	> 420 nm	707		7 ⁹
NU-100	Pt 1wt%	TEOA	> 400 nm	610	--	8 ¹⁰

Comment 3:

The authors might want to include a larger version of Fig. 3B in the supporting information as it's hard to read of. Similarly, they might want to put a table containing the underlying data in the supporting information.

Reply:

We have enlarged the image of long-term hydrogen production (Figure R2b) and added the Table R2 in the electronic supplementary information as Table S2 to make Figure R2b easier to read.

Figure R2 (Figure 3 in manuscript). (a) Time dependent hydrogen evolution for PY-DHBD-COF with different Pt loading amount. (10 mg catalyst was dispersed in 100 mL water, 10 mM ascorbic acid as electron donor, H₂PtCl₆ (0.376 g Pt L⁻¹) as Pt precursor, 300 W Xe lamp, $\lambda > 420$ nm). (b) Long-term hydrogen production for 3 wt% Pt loaded PY-DHBD-COF (detailed data are listed in Table S2).

Table R2 (Table S2 in Supporting information). Long-term hydrogen production for 3 wt% Pt loaded PY-DHBD-COF.

Term 1		Term 2		Term 3	
Time (h)	H ₂ Envoluation (mmol g ⁻¹)	Time (h)	H ₂ Envoluation (mmol g ⁻¹)	Time (h)	H ₂ Envoluation (mmol g ⁻¹)
0.0	0.0	7.5	0.0	12.5	0.0
0.5	52.1	8.0	51.5	13.0	57.9
1.0	76.0	8.5	78.4	13.5	90.9
1.5	87.5	9.0	104.3	14.0	120.2
2.0	106.3	9.5	128.6	14.5	147.6
2.5	120.8	10.0	151.7	15.0	172.1
3.0	134.3	10.5	172.3	15.5	194.8
3.5	146.9	11.0	191.3	16.0	210.2
4.0	158.6	11.5	206.3	16.5	218.9
4.5	169.4	12.0	211.6	17.0	219.7
5.0	178.6	12.5	211.7	17.5	216.2
5.5	187.8				
6.0	195.6				
6.5	203.0				
7.0	210.2				

7.5	215.7		
-----	-------	--	--

Term 4		Term 5		Term 6	
Time (h)	H ₂ Evoluation (mmol g ⁻¹)	Time (h)	H ₂ Evoluation (mmol g ⁻¹)	Time (h)	H ₂ Evoluation (mmol g ⁻¹)
17.5	0.0	22.5	0.0	27.5	0.0
18.0	60.3	23.0	57.0	28.0	54.4
18.5	90.3	23.5	92.4	28.5	84.7
19.0	120.1	24.0	121.8	29.0	111.3
19.5	146.8	24.5	148.4	29.5	135.1
20.0	172.0	25.0	171.1	30.0	153.0
20.5	193.0	25.5	191.1	30.5	175.8
21.0	212.8	26.0	209.1	31.0	188.5
21.5	225.5	26.5	222.6	31.5	212.2
22.0	227.0	27.0	221.6	32.0	212.3
22.5	226.2	27.5	221.2	32.5	212.4

Term 7		Term 8		Term 9	
Time (h)	H ₂ Evoluation (mmol g ⁻¹)	Time (h)	H ₂ Evoluation (mmol g ⁻¹)	Time (h)	H ₂ Evoluation (mmol g ⁻¹)
32.5	0.0	37.5	0.0	42.5	0.0
33.0	55.9	38.0	52.5	43.0	48.3
33.5	90.6	38.5	81.4	43.5	74.7
34.0	122.5	39.0	117.4	44.0	100.2
34.5	148.3	39.5	138.6	44.5	124.0
35.0	171.3	40.0	157.1	45.0	146.4
35.5	190.5	40.5	174.9	45.5	166.4
36.0	202.3	41.0	188.2	46.0	186.2
36.5	207.0	41.5	210.0	46.5	203.3
37.0	207.7	42.0	216.0	47.0	206.5
37.5	207.9	42.5	216.1	47.5	207.3

Term 10		Term 11		Term 12	
Time (h)	H ₂ Evoluation (mmol g ⁻¹)	Time (h)	H ₂ Evoluation (mmol g ⁻¹)	Time (h)	H ₂ Evoluation (mmol g ⁻¹)
47.5	0.0	52.5	0.0	57.5	0.0

48.0	50.9	53.0	59.2	58.0	50.9
48.5	80.2	53.5	81.8	58.5	80.2
49.0	105.7	54.0	103.8	59.0	106.7
49.5	130.4	54.5	127.0	59.5	130.4
50.0	147.9	55.0	144.2	60.0	147.9
50.5	166.0	55.5	161.2	60.5	166.1
51.0	188.1	56.0	176.7	61.0	188.1
51.5	197.3	56.5	191.8	61.5	197.4
52.0	199.0	57.0	213.7	62.0	199.0
52.5	206.5	57.5	214.6	62.5	206.5

Comment 4:

I don't see the justification for the normalisation of the HER with respect to 1 wt% Pt in Fig. 3C obtained apparently by dividing the HER by the platinum loading. I think it's better for the authors to present the unadulterated data in Fig. 3C. Also, because this HER data has been measured on different setups I am, as discussed above, sceptical about the meaning of such a comparison.

Reply:

Thanks for the reseanable suggestion. We presented the normalisation of the HER with respect to Pt for evaluating the utilization efficiency of noble metal Pt for hydrogen evolution. According to the suggestion from you and Reviewer 1, we normalized the hydrogen evolution only by Pt to evaluate the utilization efficiency of Pt as shown in figure R2 (Figure S13 in Supporting information). What's more, the direct comparison between our results and those in other reported works was canceled as the experiments were conducted under different setups.

Figure R2 (Figure S13 in Supporting information). HER activity normalized according to the loading amount of Pt.

Comment 5:

DFT calculations of the adsorption of a species on another species really should employ an empirical dispersion correction, e.g. the Grimme D3 dispersion correction giving B3LYP+D3 in the case of B3LYP, which does not seem to be the case here. Please note, that because of the presence of Pt use of Grimme's D4 dispersion correction is probably preferred over the D3 correction but I don't know if the former has already been implemented for Gaussian.

Reply:

Thank you very much for your valuable suggestions. According to your suggestions, we have recalculated all the DFT calculations. In the new calculations, the D3 version of Grimme's dispersion has been used in all the DFT and TD-DFT calculations. As you say, the D4 dispersion is preferred over the D3 correction for Pt, while it is a pity that D4 version has not been implemented for Gaussian.

Comment 6:

I might be missing something but what happened to the hydroxyl protons in Fig. 5A and S10C, E and G? Are the authors somehow assuming that the hydroxyls are deprotonated under the conditions where Pt is deposited? This feels rather odd and doesn't seem to be discussed in the manuscript. Overall, it makes me uncertain about the accuracy and significance of the values of the binding energies reported.

Reply:

Thanks for your valuable suggestion. The hydroxyls seem to be not easy to deprotonated under the acid environment. In the new calculations, we used the non-deprotonated fragments of PY-DHBD-COF to conduct the initial adsorption process.

Comment 7:

The authors write "For the geometry optimizations and frequency calculations, we used the standard 6-31g* basis set, while single-point energy calculations were performed at each stationary point using the extended 6-31++g**" in the methods section. However, I am not aware that there is a 6-31g* or 6-31++g** basis-set for Pt, and even if there was the high-Z of Pt would probably require a relativistic ECP. Again, this makes me uncertain about the accuracy and significance of the values of the binding energies reported.

Reply:

We are sorry for the mistake. In fact, we used SDD basis set combining with SDD ECP for Pt. As described in the supporting information, "The DFT calculations was carried out using the B3LYP hybrid Exchange–Correlation (XC) functional with the D3 version of Grimme's dispersion as implemented in Gaussian 09. The ground-state geometry optimization calculation employed the 6-31G(d) basis set for C, H, O, N, Cl, the Stuttgart/Dresden (SDD) basis set and the corresponding effective core potential (ECP)

was employed for Pt. Considering the influence of anion group (PtCl_6^{2-}), 6-31G+(d) basis set was used for C, H, O, N, Cl atoms in the single point energy calculation. It noteworthy that add diffuse functions for Pt has little effect on the binding energy calculation as shown in Table S*. Thus, we still use SDD combining with corresponding ECP in the single point energy calculation. Solvent effects of water were approximated by the SMD solvation model. SMD solvent effects were considered both in single point calculations and full geometry optimizations including in the case of TD-DFT calculations.

Comment 8:

In the supporting information adsorption values for I assume, based on Fig. S10A and S10B, Pt(IV)Cl_4 and Pt(II)Cl_2 are reported, with the values quoted in the main text apparently those predicted for the case of Pt(II)Cl_2 . Why were these specific platinum complexes studied? I note that experimentally the Pt source is $\text{H}_2\text{Pt(IV)Cl}_6$, which I assume is in water present as $[\text{Pt(IV)Cl}_6]^{2-}$, which appears far removed from Pt(II)Cl_2 . I can't find any discussion justifying the choice of Pt cluster in the calculations, as a matter of fact I don't think the main text mentions the type of Pt cluster used in the calculations at all. There is also the issue that the Pt deposition chemistry takes place in water and not in the gas-phase and that really an implicit solvation model should have been used, possibly in combination with explicit water to complete the coordination sphere of platinum. Again, this makes me uncertain about the accuracy and significance of the values of the binding energies reported.

Reply:

Thank you very much for your valuable suggestion. In the new calculations, we used the precursor $[\text{Pt(IV)Cl}_6]^{2-}$ as the initial Pt cluster. According to your suggestion, SMD solvent effects were considered both in single point calculations and full geometry optimizations including in the case of TD-DFT calculations.

Comment 9:

Overall, I don't think the calculated Pt complex adsorption energies can be trusted as it stands and I remain sceptical that even if they were correct that they would necessarily support the mechanism of Pt deposition proposed by the authors over alternative deposition pathways or why Pt deposition is different for this COF relative to other COFs in the literature.

Reply:

We have recalculated all the DFT calculation according to reviewer's suggestions and we think the results necessarily support the mechanism of Pt deposition. In the new calculations, we used DFT and TD-DFT calculations to give a deep insight in the Pt deposition mechanism on the PY-DHBD-COF. We also conducted comparison calculations among the three similar structures of PY-DHBD-COF, PY-BPY-COF and PY-BP-COF. The adsorption of Pt precursor (PtCl_6^{2-}) on the COF fragments and the photogenerated charge separation were calculated. Combining the adsorption free energy calculations and hole-electron distribution analysis, Pt precursors tend to adsorbed around the hydroxyl group and imine-N site of PY-DHBD-COF spontaneously, and the photogenerated electrons favour to transfer to the same site, which is benefit to the photoreduction of Pt on this specific site (Figure R7). We optimized the geometry of the PY-DHBD-COF- PtCl_6^{2-} in its singlet excited state S1 using TD-DFT method. The full optimized configuration (Figure R8) shows that two Pt-Cl bonds are significantly elongated and one Cl atom formed a weak H-bond with the $-\text{OH}$ group. This loosed configuration is favorable to loss Cl^- and form a Pt(III)Cl_5^{2-} species and lead the following photodeposition reaction. Combining with the gradually changing morphology observed by TEM and the DFT/TD-DFT calculations, we propose that Pt atoms are likely to be first adsorbed on the hydroxyl and imine site on the structural unit of PY-DHBD-COF and further reduced into clusters and metallic particles with increasing precursor concentration during the photodeposition process. Two possible photodeposition reaction paths are calculated as shown in Figure R9.

As for PY-BPY-COF and PY-BP-COF, the combination of PtCl_6^{2-} on them are thermodynamics unsupported, which is the possible reason for the non-uniform deposition of Pt. when it comes to other COFs in the literature, the framework structures are totally different with PY-DHBD-COF in this work, and the charge distribution and adsorption behaviour of PtCl_6^{2-} on them differ from each other. So it is difficult to assess the Pt deposition on different COF structures. Moreover, we found that the deposition position

and process of Pt co-catalyst on COFs were rarely discussed in detail in other literatures, which we think be of great significance for the photocatalytic hydrogen evolution. In this work, we hope to understand this more deeply through the experiments and theoretical calculations.

The relevant results, including figures, tables and the corresponding descriptions, were added in the revised manuscript and supplementary information.

Figure R7 (Figure 5 in Manuscript). (a) The adsorption free energy of PtCl_6^{2-} on three sites of PY-DHBD-COF, PY-BPY-COF and PY-BP-COF (Insert: the most stable adsorption configuration of PY-DHBD-COF- PtCl_6^{2-}). The hole and electron distribution of S1 excited state on PY-DHBD-COF (b) and the complex PY-DHBD-COF- PtCl_6^{2-} (c).

Figure R8 (Figure S20 in Supporting information). The full optimized configurations of ground state (a) and S1 excited state (b) of PY-DHBD-COF- PtCl_6^{2-} .

Figure R9 (Figure S21 in Supporting information). Free energy diagrams of two possible photodeposition reaction paths.

Comment 10:

Why did the authors switch from B3LYP to HSEH1PBE for obtaining the molecular orbitals? Again this does not seem to be discussed in the manuscript.

Reply:

In the new calculation, all the DFT calculations including the molecular orbital analysis were conducted using the B3LYP functional. The TD-DFT calculations were performed

with PBE0-D3 method instead of B3LYP, as the PBE0 functional is more reliable for the excitation energies calculation.

Comment 11:

The authors should include all relevant DFT optimised structures in the supporting information in the form of machine-readable files, i.e. as e.g. xyz files rather than coordinated pasted in a PDF. Similarly, they should really include a table with the total energies, entropies and free energies of all species in the supporting information.

Reply:

Thank you very much for your kind suggestion. We packaged the XYZ files of all the structures involved in this work into a "Configuration.rar" file as the supporting information, and added Table S10 to list their main calculation data.

Table R3 (Table S10 in Supporting information). The electron energies¹, zero point energy, entropies² and free energies³ of all species in this work.

COFs	Complex	E_{ele} (Hartree)	ZPE (Hartree)	S (Cal/mol·K)	G (Hartree)
PY-DHBD-COF	FragH	-2491.405284	0.772007	281.147	-2490.720079
	FragH-PtCl ₆ -Site1	-5372.488898	0.780492	347.342	-5371.813398
	FragH-PtCl ₆ -Site2	-5372.482675	0.780483	351.005	-5371.808825
	FragH-PtCl ₆ -Site3	-5372.487434	0.780807	348.983	-5371.812394
	FragH-PtCl ₆ - Site1-TD-OPT ³	-5373.045082	0.780499	346.313	-5372.362381
	[FragH-PtCl ₅] ²⁻	-4912.27856	0.778526	341.242	-4911.603559
	[FragH-PtCl ₄] ²⁻	-4452.081512	0.777221	337.164	-4451.407571
	[FragH-PtCl ₃] ²⁻	-3991.826069	0.775336	331.538	-3991.152783
	[Frag-PtCl ₂] ²⁻	-3530.982974	0.762394	314.66	-3530.319203
	[Frag-PtCl] ⁻	-3070.628315	0.762492	295.203	-3069.956059
	Frag-Pt	-2610.330194	0.760441	292.796	-2609.660309
	[Frag-PtCl ₅] ³⁻	-4911.812315	0.764739	353.142	-4911.156083
	[Frag-PtCl ₄] ³⁻	-4451.584916	0.763782	333.556	-4450.923
	[Frag-PtCl ₃] ³⁻	-3991.337903	0.761105	318.787	-3990.674148
	[Frag-PtCl ₂] ³⁻	-3531.108647	0.761857	320.86	-3530.446189
	[Frag-PtCl] ²⁻	-3070.740393	0.761388	303.758	-3070.072793

PY-BPY-COF	FragH	-2373.075214	0.739592	268.176	-2372.418632
	FragH-PtCl ₆ -Site1	-5254.125172	0.74793	342.986	-5253.482314
	FragH-PtCl ₆ -Site2	-5254.123194	0.747928	338.781	-5253.478429
	FragH-PtCl ₆ -Site3	-5254.12437	0.747929	336.259	-5253.478488
PY-BP-COF	FragH	-2341.031909	0.763474	267.153	-2340.350806
	FragH-PtCl ₆ -Site1	-5222.080381	0.771582	339.928	-5221.412231
	FragH-PtCl ₆ -Site2	-5222.079512	0.771548	340.73	-5221.411554
	FragH-PtCl ₆ -Site3	-5222.07956	0.771864	340.471	-5221.411686

¹ The electron energy is calculated at the B3LYP-D3/6-31G+* (C, N, O, H, Cl) & SDD+ECP (Pt) level.

² The thermal correction to Gibbs free energy is calculated at the B3LYP-D3/6-31G* (C, N, O, H, Cl) & SDD+ECP (Pt) level.

³ The TD-DFT optimization is conducted at the TD-B3LYP-D3/6-31G* (C, N, O, H, Cl) & SDD+ECP (Pt) level, and TD-DFT single point analysis is conducted at the TD-PBE0-D3/6-31G* (C, N, O, H, Cl) & SDD+ECP (Pt) level

Reviewer #3:

In this work titled “In situ photodeposition of platinum clusters on a covalent organic framework for light driven hydrogen production”, the authors report the preparation of photocatalytic systems based on covalent organic frameworks and photodeposited platinum clusters as cocatalysts for the hydrogen evolution reaction. Emphasis is given to the successful photodeposition of well-dispersed platinum clusters on the PY-DHBD-COF photocatalyst containing hydroxyl and imine functional groups, which resulted in one of the highest hydrogen evolution rates for this type of materials. I find this research study very interesting and I thus recommend this manuscript for publication in Nature Communications after the following issues are addressed:

We are grateful to the reviewer’s positive comments on our work. We have carried out supplementary experiments and made modifications according to the constructive suggestions. Detailed responses are provided as follows.

Comment 1:

The authors compare MOFs and COFs as photocatalysts, stating that the former class offers very limited stability as opposed to COFs and their application in photocatalysis is limited. However, both statements are not necessarily true as there is a large number of MOFs that are very stable photocatalysts (e.g., MIL-125, UIO-66, MIL-167, AI-PMOF to name a few, Energy Environ. Sci., 2015, 8, 364, ACS Appl. Mater. Interfaces 2021, 13, 12, 14239–14247), while COFs can suffer from stability especially under elevated temperature. I thus believe that this part of the introduction should be adjusted so as to highlight the real advantage of COF photocatalysts.

Reply:

We apologize for the inappropriate comparison between COFs and MOFs. To highlight the real advantage of COF photocatalysts, we adjusted the first paragraph of Introduction and described the properties of COFs more detailly, and deleted the unnecessary introduction of MOF materials as our work is totally based on a COF photocatalysts. Finally, the revised part is “Among the various photocatalysts investigated so far^{4,5}, porous

materials with nanoscale pore structure and ultra-high surface area have been favoured by researchers by providing larger active spaces for the adsorption of reactants and the subsequent catalytic reactions⁶⁻⁹. Covalent organic frameworks (COFs), a class of crystalline porous polymers built by covalent bonds between various ligands, have shown great advantages in photocatalytic research in recent years¹⁰. The entire covalent bond linkage makes COFs possess excellent chemical stabilities, especially in imine-linked and other nitrogen-containing COFs¹¹. As it is possible to atomically precise integration of organic units to create predesigned and various skeletons, COFs have unlimited chemical adjustability and their optical and electrical properties such as light capture, charge separation and charge transport can be modulated. The π -conjugated structure of COFs both in plane and between layers is beneficial to increase the mobility of charge carriers¹². Moreover, compared with traditional molecular catalysts, COFs lock the photoactive building units in a rigid structure to prevent photocorrosion and improve the life of the excited state¹³ as highlighted in the revised manuscript.

Comment 2:

The authors state: ‘.. compared with traditional molecular catalysts, COFs lock the photoactive building units in a rigid structure to prevent photocorrosion and improve the life of the excited state’. Please provide references of research studies in the literature supporting this argument.

Reply:

Our statement could be supported by the literatures reported by S. Yang et al. (J. Am. Chem. Soc., 2018, 140, 14614-14618) and by K. Gottschling et al. (J. Am. Chem. Soc., 2020, 142, 12146-12156), in which the molecular catalysts $\text{Re}(\text{bpy})(\text{CO})_3\text{Cl}$ and cobaloxime were locked on the periodic structures of COFs, and the enhanced electron transfer from photosensitizer to the active site retarded the deactivation of the molecular catalysts and increased the lifetime of the excited state.

Comment 3:

The in-plane pore channels of the proposed AA-stacked PY-DHBD-COF demonstrate a size of around 2.5 nm. Such value could allow the presence of Pt clusters within these pore channels, which the authors do not sufficiently describe. I thus recommend a description of the potential location of the Pt clusters. Carrying out N₂ adsorption-desorption experiments for the COFs with photodeposited Pt clusters would show how much the porosity is compromised due to the presence of Pt clusters within the pores.

Reply:

Thanks for the reasonable suggestion. Accordingly, N₂ adsorption-desorption experiments were performed on 1% and 3% Pt loaded PY-DHBD-COFs, as well as the light-treated PY-DHBD-COF without Pt loading to evaluate the impact on porosity from light irradiation during the photodeposition process. Light-treated PY-DHBD-COF was obtained under the same photodeposition condition except for the existence of Pt precursor H₂PtCl₆. The results are shown in Figure R10 and Table R4. The porosity was retained on light-treated PY-DHBD-COF, indicating the structural stability under light. After 1% and 3% Pt were loaded on PY-DHBD-COF, both surface area and pore volume have decreased significantly, which might be caused by the existence of Pt inside the pore channels. Therefore, we added the sentence “Thus small Pt clusters are possible to exist in the pore channels of PY-DHBD-COF. To check the entrance of Pt clusters into the pore channels, N₂ adsorption-desorption experiments were performed on 1 wt% Pt-PY-DHBD-COF, as well as a light-treated PY-DHBD-COF obtained under the same photodeposition condition except for the existence of Pt precursor H₂PtCl₆ in the solution (Figure S14 and Table S3). The porosity was retained on the light-treated PY-DHBD-COF, indicating the structural stability of PY-DHBD-COF under light irradiation during the photodeposition process. After Pt clusters were loaded, both surface area and pore volume decreased, which might be caused by the entrance of Pt clusters in the pore channels” in the revised manuscript and added Figure R10 and Table R4 in the electronic supplementary information as Figure S14 and Table S3, respectively.

Figure R10 (Figure S14 in Supporting information). N₂ adsorption isotherms of light-treated PY-DHBD-COF (a) and 1%wt Pt-PY-DHBD-COF (b).

Table R4 (Table S3 in Supporting information). Porosity parameters of PY-DHBD-COF and 1wt%-Pt-DHBD-COF

Samples	Surface area (m ² g ⁻¹)	Pore volume (cm ³ g ⁻¹)
PY-DHBD-COF _(original)	1893	0.918
PY-DHBD-COF _(light-treated)	1603	0.874
1wt%-Pt-PY-DHBD-COF	1106	0.663

Comment 4:

After the photocatalytic recycling experiments the preservation of the crystallinity was evaluated through PXRD. However, PXRD detects only significant changes in crystallinity (Angew. Chem. 2019, 131, 18007 – 18012). To ensure that the crystallinity and porosity of the COF is retained after the experiments, please provide N₂ sorption isotherms of the photocatalyst after the long-term photocatalytic test.

Reply:

Thanks for the reasonable suggestion. The N₂ adsorption-desorption isotherms of 3wt%Pt-PY-DHBD-COF before and after the long-term photocatalytic test were conducted and the results are shown in Figure R11 and Table R5. 3wt%Pt-PY-DHBD-COF still possessed the porous structure ($S_{\text{BET}} = 900 \text{ m}^2 \text{ g}^{-1}$ and Pore volume = $0.568 \text{ cm}^3 \text{ g}^{-1}$). However, the surface area and pore volume decreased when compared with those before long-term reaction ($S_{\text{BET}} = 571.4 \text{ m}^2 \text{ g}^{-1}$ and Pore volume = $0.416 \text{ cm}^3 \text{ g}^{-1}$), which demonstrates that the pore structure of PY-DHBD-COF is partially destroyed with the increase of reaction time. We added the Fig. R11 and Table R5 in the electronic supplementary information and revised the sentence to “XRD and N₂ absorption-desorption isotherms of the 3 wt% Pt-loaded PY-DHBD-COF before and after the long-term photocatalytic reaction were conducted to check the structure stability. The crystallinity of COF was still retained as no obvious change was found in the XRD pattern (Figure S4), but the pore structure was partially destroyed as the surface area and pore volume decreased (Figure S5 and Table S3).” in the revised manuscript.

Figure R11 (Figure S5 in Supporting information). N₂ sorption isotherms of 3wt%Pt-PY-DHBD-COF before and after the long-term photocatalytic test.

Table R5 (Table S3 in Supporting information). Porosity parameters of 3wt%-Pt-DHBD-COF before and after the long-term photocatalytic reaction.

Samples	Surface area ($\text{m}^2 \text{ g}^{-1}$)	Pore volume ($\text{cm}^3 \text{ g}^{-1}$)
3wt%-Pt-PY-DHBD-COF _(before)	900	0.568
3wt%-Pt-PY-DHBD-COF _(after)	571.4	0.416

Comment 5:

Metrics based on the incident or absorbed radiation (e.g., Apparent quantum yield/ quantum efficiency) are more appropriate to evaluate the photocatalytic activity and to facilitate the comparison with other photocatalysts using different set-up. Please provide the Apparent Quantum Yield (at wavelength >420 nm) or Quantum efficiency of the best photocatalyst.

Reply:

Thanks for the reasonable suggestion, which is also mentioned by the other two reviewers. Accordingly, we compared the external quantum efficiencies to evaluate the photocatalytic activity as shown in Table R1.

Table R1 (Table S4 in Supporting information). The photocatalytic hydrogen evolution performance comparison of PY-DHBD-COF with other representative COF and MOF based photocatalysts.

Entry	Co-catalyst	Sacrificial reagent	Light Source	HER ($\mu\text{mol g}^{-1} \text{h}^{-1}$)	EQE (at 420 nm) (%)	Ref.
PY-DHBD-COF	Pt 0.5wt%	Ascorbic acid	> 420 nm	16980	--	This work
PY-DHBD-COF	Pt 1wt%	Ascorbic acid	> 420 nm	42432	--	This work
PY-DHBD-COF	Pt 2wt%	Ascorbic acid	> 420 nm	56712	--	This work
PY-DHBD-COF	Pt 3wt%	Ascorbic acid	> 420 nm	71160	8.4	This work
PY-DHBD-COF	Pt 5wt%	Ascorbic acid	> 420 nm	48912	--	This work
Tp-2C/BPy2+-COF (19.10%)	Pt 3wt%	Ascorbic acid	> 420 nm	34600	6.93	1 ³

Py-CITP-BT-COF	Pt 5wt%	Ascorbic acid	> 420 nm	8875	8.45	2 ⁴
FS-COF	Pt 8wt%	Ascorbic acid	> 420 nm	10100	3.2	3 ⁵
g-C ₁₈ N ₃ -COF	Pt 3wt%	Ascorbic acid	> 420 nm	2920	4.84	4 ⁶
MIL-125/Au	Pt 0.49wt%	TEOA	> 420 nm	1743	--	5 ⁷
Uio-66-NH ₂	Pt 0.65wt%	Ascorbic acid	> 420 nm	1528	2.3	6 ⁸
MIL-125-NH ₂	Pt 0.45wt%	TEOA	> 420 nm	707		7 ⁹
NU-100	Pt 1wt%	TEOA	> 400 nm	610	--	8 ¹⁰

Comment 6:

The authors state: ‘.. the fluorescence lifetimes of the PY-DHBD-COF samples are shorter than those of the corresponding PY-BPY-COF samples (1.09 ns < 1.79 ns; 1.93 ns < 2.56 ns), indicating more efficient charge separation of the PY-DHBD-COF samples.’ However, shorter lifetimes correspond to faster recombination. What do the authors mean by this statement?

Reply:

Thanks for reviewer’s comment. After careful consideration, we think the statement that the reviewer mentioned in this comment is inappropriate and deleted it in the revised manuscript. The fluorescence lifetime provided details about charge separation efficiency of the COFs. In fact, the fluorescence lifetime data fitted into the bi-exponential equation (1):

$$I(t) = A_1 \cdot \exp(-t/\tau_1) + A_2 \cdot \exp(-t/\tau_2), \quad (1)$$

where $I(t)$ is the fluorescence intensity at a certain delay time, τ_1 and τ_2 are fluorescence lifetimes, and A_1 and A_2 are the relative weights of the decay components at $t = 0$. This indicates two relaxation pathways of the charge carriers, radiative relaxation and

nonradiative relaxation. The fluorescence lifetime is related to both two relaxation pathways as shown in equation (2):

$$\tau = (k_r + k_{nr})^{-1} \quad (2)$$

where k_r is radiation rate constant and k_{nr} is nonradiative rate constant. Therefore, a shorter lifetime corresponds to faster recombination or faster nonradiative relaxation. For a photocatalyst, efficient charge separation and transfer lead to increased nonradiative relaxation pathway, which will lead to short fluorescence lifetime. However, it is difficult to estimate the contributing proportion of radiative relaxation and nonradiative relaxation pathways according to the fluorescence lifetimes of two different materials, such as PY-DHBD-COF and PY-BPY-COF. Therefore, it is inappropriate to compare the lifetimes of these two COF materials directly.

As for one COF material before and after Pt loading, the quenched fluorescence is mainly caused by the increased nonradiation relaxation pathway due to the electron transfer from COF to Pt. Hence, the shorter lifetime of Pt loaded COF correspond to increased k_{nr} and more efficient charge separation.

Comment 7:

At the transient photoluminescent experiments (TPL), the recorded PL emission was at 620 nm, which corresponds to the highest PL intensity of PY-BPY-COF, but not to that of PY-DHBD-COF. Therefore, I believe that the TPL of PY-DHBD-COF at its maximum PL intensity (around 570 nm) should be provided for a valid comparison.

Reply:

According to reviewer's suggestion, the transient fluorescence spectra of PY-DHBD-COF and 1wt%Pt-PY-DHBD-COF were measured at 560 nm (corresponding to the highest PL intensity of PY-DHBD-COF) and the results are shown in Figure R12. The fluorescence lifetime of 1wt%Pt-PY-DHBD-COF is shorter than that of PY-DHBD-COF, indicating more efficient charge separation in 1wt%Pt-PY-DHBD-COF. finally, we added Figure R12 in the revised manuscript as Figure 6c.

Figure R12 (Figure 6c in Manuscript). Fluorescence lifetime decay of PY-DHBD-COF and 1wt%-Pt-PY-DHBD-COF. Samples were excited with a $\lambda_{\text{ex}} = 374$ nm laser and emission was measured at 560 nm.

REVIEWER COMMENTS

Reviewer #2 (Remarks to the Author):

I think the changes made by the authors in the revised manuscript are sensible and have improved the manuscript. I also feel that the authors have generally addressed my comments on the previous version of the manuscript well.

After reading the revised manuscript I have three, hopefully final, comments:

1) Assuming Fig. S16 shows indeed the optimised structures, it feels slightly unexpected that the free energy of adsorption for sites 2 and 3 for PY-DBHD-COF are so similar to that predicted for site 1 for the same COF fragment, where the OH directly interacts with the Pt precursor, and are so more negative than for say sites 1 or 3 for PY-BPY-COF. I would not have expected that the addition of OH groups would give rise to a difference in adsorption (free) energy when not directly involved in the bonding and when the Pt precursor is located on the other side of the linker.

2) It's unclear to me what the authors mean when they say they plot "the hole and electron distribution of S1 excited state" in Fig. 5B/C and Figs. S18 and S19. The excited state should have both a hole and an electron component, but the figures only seem to have two colour "shadings". Also, I don't think there is a general method for plotting the hole and electron distribution of an excited state so the authors should explain more clearly what they have done. Finally, they should really explain in the captions what the numbers underneath the figures (S & D) signify.

3) I have no clue what the authors mean with electron-hole theory and/or how they calculate the S & D values. I would expect both references and an explanation of how this is conceptually calculated and how this is done in practice (i.e. what code what used and which settings). MultiWFN and VMD are mentioned but both codes, and especially VMD, can calculate/visualise/analyse many things so the authors need to be clearer.

Reviewer #3 (Remarks to the Author):

The authors have addressed all my comments, and therefore I recommend this manuscript for publication after the following 2 minor issues (related to my previous comments, 4 and 5) are addressed:

Comment 1 (related to old comment 4): Thank you for taking my comment into consideration and performing the N₂ adsorption-desorption experiments. The decrease in the surface area and pore volume after the long-term photocatalysis indicates that there were losses in the crystallinity. Please rephrase the following (newly-added) sentence in the manuscript accordingly: 'The crystallinity of COF was still retained as no obvious change was found in the XRD.'

Comment 2 (related to old comment 5): The authors mention that they measured the external quantum efficiency at 420 nm. However, by definition quantum efficiency refers to the use of wavelength in a range λ_1 – λ_2 (not the use of monochromatic radiation which is involved in the Apparent Quantum Yield term), and is based on the absorbed radiation (not on incident radiation as in the case of photonic efficiency). The authors most probably calculated the Apparent Quantum Yield value at 420 nm, however this is not with certainty, as no information on how this value was calculated is reported. Please provide information about the methodology used to calculate this value, and properly define the metric used to evaluate the efficiency, based on IUPAC recommendations and/or literature focused on metrics used for reporting photocatalytic activities (Pure Appl. Chem., Vol. 83, No. 4, pp. 931–1014, 2011 and/or Chem. Mater. 2017, 29, 158–167).

Point-by-point response to the reviewers' comments on NCOMMS-21-21627A

Reviewer #2: I think the changes made by the authors in the revised manuscript are sensible and have improved the manuscript. I also feel that the authors have generally addressed my comments on the previous version of the manuscript well. After reading the revised manuscript I have three, hopefully final, comments.

We are grateful to the reviewer's further comments on our work. Detailed responses are provided as follows.

Comment 1:

Assuming Fig. S16 shows indeed the optimised structures, it feels slightly unexpected that the free energy of adsorption for sites 2 and 3 for PY-DHBD-COF are so similar to that predicted for site 1 for the same COF fragment, where the OH directly interacts with the Pt precursor, and are so more negative than for say sites 1 or 3 for PY-BPY-COF. I would not have expected that the addition of OH groups would give rise to a difference in adsorption (free) energy when not directly involved in the bonding and when the Pt precursor is located on the other side of the linker.

Thank you very much for your professional suggestion. We rechecked our calculation results carefully and found that we made a mistake in the adsorption energy calculation of PtCl_6^{2-} on PY-DHBD-COF by using the unoptimized structure PY-DHBD-COF in the adsorption energy calculation which caused the abnormal adsorption energy difference. According to your suggestion, we rechecked the calculation method and recalculated the adsorption energy of PtCl_6^{2-} on PY-DHBD-COF, PY-BPY-COF and PY-BP-COF, and the results are shown in Table S5 and Figure 5a.

Table S5. Adsorption energy (ΔE_{ads}) of PtCl_6^{2-} on COFs.

COFs	Configurations	ΔE_{ads} (Kcal/mol)
PY-DHBD-COF	Site1	-8.96
	Site2	-5.25
	Site3	-8.15
PY-BPY-COF	Site1	-6.72
	Site2	-5.41
	Site3	-6.17
PY-BP-COF	Site1	-5.83
	Site2	-5.40
	Site3	-5.27

Figure 5. (a) The adsorption energy of PtCl_6^{2-} on three sites of PY-DHBD-COF, PY-BPY-COF and PY-BP-COF (Insert: the most stable adsorption configuration of PY-DHBD-COF- PtCl_6^{2-}). The hole (lime) and electron (violet) distribution of S1 excited state on PY-DHBD-COF (b) and the complex PY-DHBD-COF- PtCl_6^{2-} (c). S is the overlap integral of hole-electron distribution and D means the distance between centroid of hole and electron.

The lowest adsorption energies of the PtCl_6^{2-} (ΔE_{ads}) on PY-DHBD-COF, PY-BPY-COF and PY-BP-COF are -8.96, -6.72 and -5.83 kcal/mol, respectively. Therefore, the lowest adsorption energy of PtCl_6^{2-} on PY-DHBD-COF is 2.24 Kcal higher than that on PY-BPY-COF and 3.13 Kcal/mol higher than that on PY-BP-COF. On the basis of the adsorption energies, the adsorption selectivity can be estimated through the equation $\text{Sel}_A/\text{Sel}_B = \exp\left[\left(\Delta E_{\text{ads}}^B - \Delta E_{\text{ads}}^A\right)/RT\right]$, and it was found that the adsorption selectivity of PtCl_6^{2-} on PY-DHBD-COF is 43.8 times that of PY-BPY-COF and 196.6 times that of PY-

BP-COF. The significant difference in adsorption energy and adsorption selectivity may lead to the most stable adsorption of PtCl_6^{2-} on the site 1 of PY-DHBD-COF.

For further understanding the influence caused by addition of -OH groups, the electrostatic potential (ESP) analysis of fragments PY-DHBD-COF, PY-BPY-COF and PY-BP-COF are conducted. The ESP on vdW surface is often investigated because it is very closely related to intermolecular interactions (*Int. J. Quant. Chem.*, 2017, 117, e25443; *J. Chem. Phys.*, 2018, 148, 194106; *Phys. Chem. Chem. Phys.*, 2021, 23, 20323-20328; *Carbon*, 2021, 171, 514-523). The ESP analysis (Figure R1) shows that the orders of molecular polarity index and the positive surface electrostatic potential average value are PY-DHBD-COF > PY-BPY-COF > PY-BP-COF. Since PtCl_6^{2-} is an anion, the greater the polarity of the framework and the more positive surface electrostatic potential average value, the stronger the interaction, which is consistent with the calculation results of the order of adsorption energy PY-DHBD-COF > PY-BPY-COF > PY-BP-COF. As a result, the introduction of -OH groups will affect the molecular polarity and the distribution of electrostatic potential, thus the adsorption energy of Pt precursor will be affected even if -OH group is not directly involved in the bonding.

Figure R1. Electrostatic potential (ESP) mapped van der Waals surface (i.e. = 0.001 a.u. isosurface) of PY-DHBD-COF (a), PY-BPY-COF (b) and PY-BP-COF (c). The redder the color, the more positive the ESP. (d) Molecular polarity index (MPI) and positive surface electrostatic potential average value (PAV) of the fragments of PY-DHBD-COF, PY-BPY-COF and PY-BP-COF.

Comment 2:

It's unclear to me what the authors mean when they say they plot "the hole and electron distribution of S1 excited state" in Fig. 5B/C and Figs. S18 and S19. The excited state should have both a hole and an electron component, but the figures only seem to have two colour "shadings". Also, I don't think there is a general method for plotting the hole and electron distribution of an excited state so the authors should explain more clearly what they have done. Finally, they should really explain in the captions what the numbers underneath the figures (S & D) signify.

Thank you for your kind suggestion. We rewrite the caption of Fig. 5B/C, Figs. S18 and S19. In Fig. 5B/C, Figs. S18 and S19, the hole and electron distribution are represented in lime and violet, respectively. The hole and electron distribution are calculated with equation R1-R6 using the Multiwfn code:

$$\rho^{\text{hole}}(r) = \rho_{\text{loc}}^{\text{hole}}(r) + \rho_{\text{cross}}^{\text{hole}}(r) \quad (\text{R1})$$

$$\rho_{\text{loc}}^{\text{hole}}(r) = \sum_{i \rightarrow a} (w_i^a)^2 \varphi_i \varphi_i - \sum_{i \leftarrow a} (w_i^a)^2 \varphi_i \varphi_i \quad (\text{R2})$$

$$\rho_{\text{cross}}^{\text{hole}}(r) = \sum_{i \rightarrow a} \sum_{j \neq i \rightarrow a} w_i^a w_j^a \varphi_i \varphi_j - \sum_{i \leftarrow a} \sum_{j \neq i \leftarrow a} w_i^a w_j^a \varphi_i \varphi_j \quad (\text{R3})$$

$$\rho^{\text{ele}}(r) = \rho_{\text{loc}}^{\text{ele}}(r) + \rho_{\text{cross}}^{\text{ele}}(r) \quad (\text{R4})$$

$$\rho_{\text{loc}}^{\text{ele}}(r) = \sum_{i \rightarrow a} (w_i^a)^2 \varphi_a \varphi_a - \sum_{i \leftarrow a} (w_i^a)^2 \varphi_a \varphi_a \quad (\text{R5})$$

$$\rho_{\text{cross}}^{\text{ele}}(r) = \sum_{i \rightarrow a} \sum_{i \rightarrow b \neq a} w_i^a w_i^b \varphi_a \varphi_a - \sum_{i \leftarrow a} \sum_{i \leftarrow b \neq a} w_i^a w_i^b \varphi_a \varphi_a \quad (\text{R6})$$

where $\rho^{\text{hole}}(r)$ and $\rho^{\text{ele}}(r)$ stand for the density distribution of hole and electron, respectively, r is the distance, φ is the orbital wave function, φ_i or φ_j is the occupied orbital, and φ_a or φ_b is the unoccupied orbital. Therefore, $i \rightarrow a$ represents excitation configuration, $i \leftarrow a$ represents de excitation configuration. Hole distribution and electron distribution are divided into local term and cross term. The local term is generally dominant, reflecting the contribution of the configuration function itself, and the cross term can not be ignored, otherwise the quantification is inaccurate, which reflects the influence of the coupling between the configuration functions on the hole and electron distribution. The calculation method was written in the supporting information files.

Comment 3:

I have no clue what the authors mean with electron-hole theory and/or how they calculate the S & D values. I would expect both references and an explanation of how this is conceptually calculated and how this is done in practice (i.e. what code what used and which settings). MultiWFN and VMD are mentioned but both codes, and especially VMD, can calculate/visualise/analyse many things to the authors need to be clearer.

Thank you for your kind suggestion. S stands for the overlap integral of hole-electron distribution and D reflects the distance between centroid of hole and electron. The S and D are calculated with equation R7-R8 using Multiwfn code:

$$S = \int \min[\rho^{\text{hole}}(r) + \rho^{\text{ele}}(r)] dr \quad (\text{R7}),$$

$$D = \sqrt{|X_{\text{ele}} - X_{\text{hole}}|^2 + |Y_{\text{ele}} - Y_{\text{hole}}|^2 + |Z_{\text{ele}} - Z_{\text{hole}}|^2} \quad (\text{R8}).$$

where $\rho^{\text{hole}}(r)$ and $\rho^{\text{ele}}(r)$ stand for the density distribution of hole and electron, respectively, X_{hole} refers to the X coordinate of centroid of hole, which can be obtained through multiplying the ρ^{hole} function by the X coordinate variable and integration in the whole space. The relevant references are: *Carbon*, 165, 461-467 (2020); *Nat. Mater.*, 19, 1332-1338 (2020); *J. Phys. Chem. Lett.*, 9, 4857 (2018); *J. Phys. Chem. C*, 121, 8091 (2017); *J. Mater. Chem. C*, 5, 5214 (2017); *Angew. Chem. Int. Ed.*, 60, 2-10, (2021). The detailed settings and process can be found in part 4.18 "Electron excitation analysis" in the manual of Multiwfn code (Tian Lu, Multiwfn Manual, version 3.8, Section 4.18, available at http://sobereva.com/multiwfn/misc/Multiwfn_3.8.pdf)

Reviewer #3: The authors have addressed all my comments, and therefore I recommend this manuscript for publication after the following 2 minor issues (related to my previous comments, 4 and 5) are addressed.

We are grateful to the reviewer's further comments on our work. Detailed responses are provided as follows.

Comment 1 (related to old comment 4):

Thank you for taking my comment into consideration and performing the N₂ adsorption-desorption experiments. The decrease in the surface area and pore volume after the long-term photocatalysis indicates that there were losses in the crystallinity. Please rephrase the following (newly-added) sentence in the manuscript accordingly: 'The crystallinity of COF was still retained as no obvious change was found in the XRD.'

Reply:

After the long-term photocatalysis, there was no obvious change found in the XRD patterns, but the surface area and pore volume decreased based on the results of N₂ sorption experiment. As the reviewer mentioned, PXRD detects only significant changes in crystallinity, while N₂ sorption isotherm enables the detection of microscopic changes in the pore structure. According to the reviewer's comment, we rephrase the sentence into : "No obvious change was found in the XRD pattern (Figure S4), whereas the surface area and pore volume decreased based on the results of N₂ sorption experiment (Figure S5 and Table S3), which indicates that there were losses in the crystallinity of the PY-DHBD-COF after the long-term photocatalytic reaction."

Comment 2 (related to old comment 5):

The authors mention that they measured the external quantum efficiency at 420 nm. However, by definition quantum efficiency refers to the use of wavelength in a range λ_1 – λ_2 (not the use of monochromatic radiation which is involved in the Apparent Quantum Yield term), and is based on the absorbed radiation (not on incident radiation as in the case of photonic efficiency). The authors most probably calculated the Apparent Quantum

Yield value at 420 nm, however this is not with certainty, as no information on how this value was calculated is reported. Please provide information about the methodology used to calculate this value, and properly define the metric used to evaluate the efficiency, based on IUPAC recommendations and/or literature focused on metrics used for reporting photocatalytic activities (Pure Appl. Chem., Vol. 83, No. 4, pp. 931–1014, 2011 and/or Chem. Mater. 2017, 29, 158–167).

Reply:

Thanks for the kind suggestion. The external quantum efficiency (EQE) we mentioned in the manuscript is indeed the Apparent Quantum Yield (AQY) the reviewer mentioned in this comment. In fact, in the literatures of photocatalysis research, AQY is also called EQE or the apparent quantum efficiency (AQE) (*Joule*, 2021, 5, 344-359; *Nature*, 581, 411-414; *Angew. Chem. Int. Ed.*, 2021, 60, 2-10). Although the names are different, the real meanings are same, namely, the number of reacted electrons occurring per incident photon by the system at a specified wavelength. Besides, another quantum efficiency called internal quantum efficiency (IQE), meaning the number of reacted electrons occurring per absorbed photon, is a more accurate and meaningful parameter to evaluate the performance of photocatalyst. However, it is difficult to determine the number of photons absorbed by particulate photocatalysts because of the scattering of photons by the photocatalyst particles. Therefore, AQY (or AQE/EQE) was commonly measured in the research of the powder photocatalysts.

According to the reviewer's comment, we modified the name of EQE into AQY, gave a simple discription of its defination, and added the information about the methodology used to calculate AQY in the revised supporting information. The relevant sentences are "To further evaluate the performance of the photocatalyst, the apparent quantum yield (AQY), defined as the number of reacted electrons occurring per incident photon by the system at a specified wavelength²⁹, was measured at 420 nm for 3 wt% Pt-PY-DHBD-COF" in the revised manuscript and "The AQY measurement was conducted in the same reaction system as other photocatalytic reactions, otherwise the xenon lamp was equipped with a band-pass filter with central wavelength of 420 nm and full-width at half-maximum (FWHM) of ~10 nm. The number of photons reaching the solution was measured using a calibrated Si photodiode. For full absorption of the incident photons, 50 mg 3% Pt-PY-DHBD-COF

was used as photocatalyst in the AQY measurement. The AQY value was calculated according to the following equation:

$$\text{AQY(\%)} = (2R(\text{H}_2) / I) \times 100$$

where $R(\text{H}_2)$ and I denote the evolution rate of H_2 in the initial one hour irradiation and the number of photons reaching the surface of the reaction solution per hour, respectively. The total number of incident photons per hour were measured to be $4.3 \times 10^{19} \text{ h}^{-1}$. The H_2 evolution rate was $3 \mu\text{mol h}^{-1}$ (this is $1.806 \times 10^{18} \text{ h}^{-1}$)." in the revised supporting information.

REVIEWERS' COMMENTS

Reviewer #2 (Remarks to the Author):

I am happy with the changes made, some hopefully now really final comments:

- 1) Why use kcal/mol as unit of the predicted (free) energies rather than the SI kJ/mol?
- 2) The authors should really also define the w s in equations S4-S8. I assume these are the coefficients for the decomposition of the excited state in terms of orbital excitations.
- 3) I am confused by the form of equation S9. I would naively have expected an overlap integral to be a function of the product of the hole and electron density rather than a sum. I also don't know what is the meaning of "min" in the integral.
- 4) It's great that the authors have now added references to their methodology used to analyse the excited states but I don't think any of those are where the methodology was first derived/proposed.

Point-by-point response to the reviewers' comments on NCOMMS-21-21627B

Reviewer #2 (Remarks to the Author):

I am happy with the changes made, some hopefully now really final comments:

1) Why use kcal/mol as unit of the predicted (free) energies rather than the SI kJ/mol?

Reply:

Thank you very much for your kind suggestion. We used Kcal/mol as the unit of the predicted energies both in the main text and SI.

2) The authors should really also define the w s in equations S4-S8. I assume these are the coefficients for the decomposition of the excited state in terms of orbital excitations.

Reply:

Thank you very much for your kind suggestion. Accordingly, we defined the w and w' in the text of supporting information. The w and w' correspond to configuration coefficient of excitation and de-excitation, respectively.

3) I am confused by the form of equation S9. I would naively have expected an overlap integral to be a function of the product of the hole and electron density rather than a sum. I also don't know what is the meaning of "min" in the integral.

Reply:

Thank you very much for your professional suggestion. Equation S9 is wrong. We corrected it as:

$$S = \int \min[\rho^{\text{hole}}(r), \rho^{\text{ele}}(r)] dr$$

where 'min' means the minimum of $\rho^{\text{hole}}(r)$ and $\rho^{\text{ele}}(r)$.

4) It's great that the authors have now added references to their methodology used to analyse the excited states but I don't think any of those are where the methodology was first derived/proposed.

Reply:

Thank you very much for your kind suggestion. We used electron-hole analysis method inserted in the Multiwfn software to do the excited states analysis. We cited the references (J. Comput. Chem. 33, 580-592 (2012), Carbon, 165, 461-467 (2020), Tian Lu, Multiwfn Manual, version 3.8, available at http://sobereva.com/multiwfn/misc/Multiwfn_3.8.pdf) in our manuscript as the authors of the Multiwfn asked. On the other hand, we conducted literature research of electron-hole distribution analysis and didn't find much more original literatures. We appreciate it very much if you point out the original reference and we will cite it in our manuscript as well.